# Relationship of Work-Related Stress and Offline Social Leisure on Political Participation of Voters in the United States

## Oldřich Šubrt 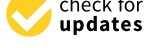

Faculty of Law, University of Amsterdam, 1000 BA Amsterdam, The Netherlands; oldasubrt@protonmail.com

**Abstract:** In the United States (US), citizens' political participation is 15%. Contemporary psychological models explaining political participation are based on education and socioeconomic status, which are unable to explain the overall low political participation figures. The study suggests a holistic approach, with two societal tendencies: increasing work-related stress and diminishing offline social leisure, together with a mediating effect of participatory efficacy to assess associations with the political participation of US voters. The quantitative correlational study uses structural equation modelling (SEM) analysis on the General Social Survey representative sample of US voters ($N = 295$, $M_{age} = 44.49$, $SD = 13.43$), controlled for education and socioeconomic status. Work-related stress was not significantly associated with political participation ($\beta = 0.08$, $p = 0.09$). Offline social leisure was positively associated with political participation ($\beta = 0.28$, $p < 0.001$). The mediating effect of participatory efficacy on the relationship between offline social leisure and political participation was positive and significant ($\beta = 0.05$, $p < 0.001$). Additional analyses, regression and SEM on the European Social Survey sample ($N = 27,604$) boosted internal and external validity. Results indicate that offline social leisure is more predictive than education and socioeconomic status, showing that examining societal trends leads to a better understanding of political participation.

**Keywords:** political participation; work-related stress; offline social leisure; participatory efficacy; resource theory of political participation; critical theory; neoliberal society; American workforce; United States electorate

## 1. Introduction

In most democratic theories, citizens are the backbone of the political process when initiating change and checking those in power (Cohen et al. 2001; Smets and van Ham 2013). With the record voter turnout (61%) in the last United States (US) presidential elections, some can infer that the US government has a strong mandate. However, most Americans disliked both candidates and voted based on the felt animosity toward the opposing candidate (Russonello 2020). Views against the government are further accented in the American society, as only 20% of Americans trust the federal government, 30% trust its transparency, and over 60% of American citizens want a fundamental change of the governmental composition (Pew Research Center 2018). With those data, it is puzzling why the majority refrains from active participation beyond electoral participation.

Only 15% of Americans actively participated in politically-related activities, such as publicly supporting or donating to a cause, contacting public officials, or attending a political event (Pew Research Center 2018). To decipher this inactivity paradox, the contemporary psychological models of political participation use social status and education as their predicting variables (Barrett and Brunton-Smith 2014; Cohen et al. 2001).

These models show which segments of society participate more in politics. However, they cannot solve the paradox, and they even perpetuate it, as contemporary society is wealthier and more educated (Pew Research Center 2020; U.S. Census Bureau 2020); and should, in theory, be more engaged in all facets of political participation (Campbell 2013). Although this inactivity is more prevalent among the less educated, who also have lower

socioeconomic positions (Cohen et al. 2001), the models do not explain the overall low participation rates, when increased participation could lead to the wished-for structural change of the governmental setup (Pew Research Center 2018).

Even though political participation studies, in sociology and political science, focus on various explanatory variables, the current psychological models of political participation base their independent variable selection on the resource theory of political participation (Barrett and Brunton-Smith 2014; Wang et al. 2017). This theory views education and socioeconomic variables as central in explaining the differences among citizens. However, it cannot describe the macroscopic tendencies when centred around comparative levels of education and status. Unfortunately, papers dealing with political psychology overlook the holistic processes affecting political participation, neglecting psychological and sociological motivations (McClurg 2003; Schlozman et al. 2018).

This paper aims to study the entire social strata to comprehend why US political participation is only 15% when the last voter turnout for the presidential election was 61%. Therefore, the study focuses on civic participatory actions as forms of political participation (Barrett and Brunton-Smith 2014) and examines voting as a distinct form of political participation since it is the most influenced by social norms (Bäck et al. 2011).

To understand why political participation is low in neoliberal society, where the primary lens of evaluating success is through one's work (Elliott 2018), the entire working conditions and norms associated with work should be considered. Moreover, since working demands are increasing (Brady 2019) and chronic work-related stress influences 60% of Americans (Boyd 2021), this paper will examine their effects on political participation via structural equation modelling on a representative sample of American citizens conducted by National Opinion Research.

The paper aims to find answers to the relationship between work-related stress and the political participation of voters in the US (see Figure 1). The first empirical aim is to advance the significant correlational findings of the overall effects of stress on voting behaviours (Ojeda et al. 2020). This effect is theorised to be explained through the cognitive load theory, proposing that cognition is also a diminishing resource alongside time and money (Balamurugan et al. 2020). The self-development theory furthers the hypothesis, which explains why most Americans are not intrinsically motivated to participate in politics (Vansteenkiste et al. 2020).

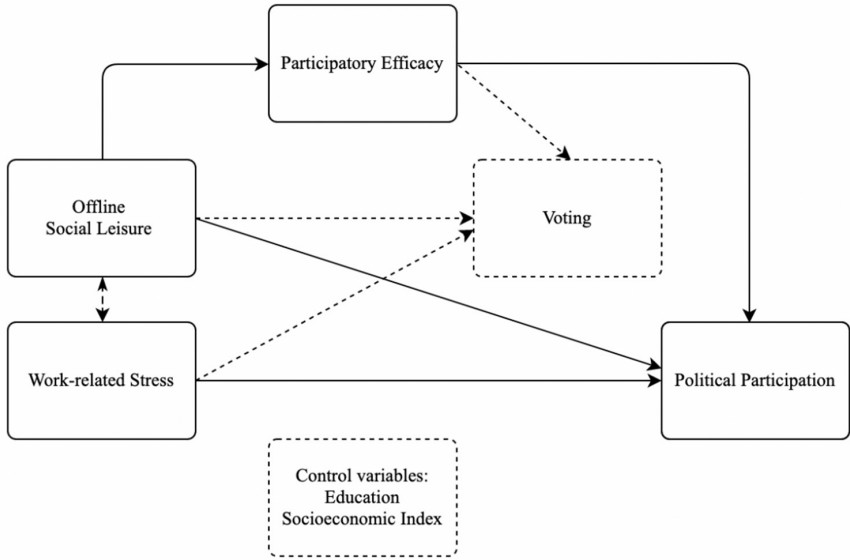

**Figure 1.** Conceptual model of the relationship between work-related stress/offline social leisure and political participation. H1 = first hypothesis. H2 = second hypothesis. H3 = third hypothesis. The political participation variable does not include voting. The full lines denote the main analysis pathways, whereas dotted lines represent the aims for exploratory analysis.

The second empirical aim explores leisure time, in which coping with work-related stress is taking effect (Meurs et al. 2010). It is empirically established that workers spend less leisure time socialising (Lee and Lee 2015). When offline socialising activities nurture close ties (Williams 2007), and since 60% of Americans are persistently feeling lonely (CIGNA 2020), it is necessary to know the relationship between offline social leisure and the political participation of voters in the US is, and if it is mediated by participatory efficacy. Social identity theory will aid us in explaining why the American population has fewer close relationships (Iacoviello and Lorenzi-Cioldi 2019), and the social identity model of collective action will show how a lack of social identity translates to political action, primarily through participatory efficacy beliefs (van Zomeren et al. 2012).

The first exploratory aim of the paper is to rule out the influence of voting preference, as neither dependent variable should be associated with voting behaviours. The second aim is to evaluate whether the model will be more predictive of political participation than education and social status alone, for which this study will control. The third aim will focus on the direct relationship between work-related stress and offline social leisure. Hence, the study aims to challenge the dominant model explaining political participation inside the discourse of political psychology.

The theoretical aim introduces a theory for the holistic model, which is not looking at the individual differences but at major trends through the extensive descriptions of the current labour force. For each research question, the essential parts necessary for understanding the model can be found in the concluding remarks of the theoretical background.

## 2. Theoretical Background

### 2.1. Work-Related Stress and Political Participation

#### 2.1.1. Neoliberalism and Its Working Conditions

Neoliberalism is defined as an ideology encouraged by the predominantly Western elite that submits to the market rule rationale (Peck and Theodore 2019), embedded in global economic freedom (Madariaga 2018), to advance their power in the form of global financial capital (Arsel et al. 2021). Notably, neoliberalism anchors its world view in trade and financial liberalisation to safeguard internal and external competitiveness (Madariaga 2018). Those presuppositions are treated as reality principles (Peck and Theodore 2019), used in political rationality and further co-produced inside the media space (Peck and Theodore 2019). Through these mechanisms, neoliberalism does not only affect society by the marketisation of all spheres of life, but it also proposes pure technocratic solutions to political challenges (Arsel et al. 2021). Hence, neoliberalism is also a process with many overlapping currents leading to progressive, regressive, authoritarian, technocratic, and reactionary policies beyond the market rationale (Peck and Theodore 2019), creating a hybrid array of neoliberal governances (Madariaga 2018).

In a neoliberal society, work is regarded as an emancipatory process of cultivating one's happiness by weighing one's contributions to the advancement of the neoliberal society (Foucault 2010). One's work-related suffering is embraced to achieve one's true potential. The citizen is seen as an entrepreneur, constantly innovating themselves while embracing the market uncertainty (Foucault 2010). Contemporary work is conceptualised as an avenue for meaning exploration and self-expression; homo economicus is no longer an exchange partner; it is instead a being of its capital (Springer et al. 2016).

The appraisal of work in the neoliberal era is aided through a meritocratic ideal corresponding to the upper-middle-class archetype, wherein one's social position corresponds to effort and inborn talent (Littler 2013). Meritocracy denotes the movement upwards toward acquiring money and status. However, with the dismantling of the welfare state, the upward moment becomes improbable for the majority of the population, which lacks the initial capital. And since one's position is judged as the consequence of one's effort, the escape from structural inequality becomes responsibilised, forming a competitive environment (Littler 2013).

The flexible, individualised members of the labour force competing against one another (Littler 2013) and globalised financial instruments have caused the hourly wage to stagnate. Prior to the 1970s, the increase in productivity was followed by a 1:1 wage increase (Przeworski 2016). However, after 1973, the American wage stagnation began. It resulted in a productivity gap in the USA (Brady 2019); the median wage rose by 6%, and the workers' productivity increased by six times between 1979 and 2013 (Rosa 2014). Moreover, 94% of new work arrangements do not offer full-time contracts, giving less protection to the upcoming workforce. Job insecurity is likely to increase because of a new rising form of technologically driven precarious work known as the gig economy (Standing 2021). Consequently, 35% of the US labour force already takes part in it (Statista 2021).

Neoliberal working conditions demand high productivity and neglect social relationships (Hefty 2020). Those conditions have produced all-time highs of self-reported work-related stress among all socioeconomic groups (Boyd 2021). Around 80% of US workers report experiences of stress, and 60% report work-related stress daily, mainly because of workload and lack of support (Boyd 2021).

All in all, work in American society has become the primary source of merit and the means of pursuing one's liberation, from which one appraises one's position in the collective (Carter 2006). With its increasing demands from the globalised competitive labour market, it is necessary to examine the effects of work-related stress on political participation.

### 2.1.2. Limitations of Current Psychological Models Predicting Political Participation

Political participation is defined as political and civic participatory actions (Barrett and Brunton-Smith 2014). The actions are voluntarily and legally enacted by ordinary citizens to influence a political outcome (Ekman and Amnå 2012; van Deth 2016). Political participatory behaviour can be expressed through conventional means, such as campaign contributions and attending political rallies, or through nonconventional means, such as protests and signing petitions (Barrett and Brunton-Smith 2014).

On the other hand, voting is a distinct form of political participation and is treated as such in this study. When compared to other forms of political participation, voting is the most influenced by social norms (Bäck et al. 2011; Galais and Blais 2014) and is habitually reinforced (Harder and Krosnick 2008)—calling into question the voluntary aspect of such action. Furthermore, the electoral system influences the number of electoral votes. Single-seat races dramatically underperform multiseat proportional races (Smith 2017). With the low electoral silence—the effect of a single vote on policy—in the United States, and varying administrative and gerrymandering processes, other forms of political participation carry higher incentives (Martinez 2010).

Even though voting is the most widespread form of political participation, it shares few commonalities with the other forms of political participation. Additionally, with the adoption of the rational choice model (Mahsud and Amin 2020) and the evidence from this paragraph, voting is among the costliest forms of political participation. Thus, following the study designs of Quintelier and Hooghe (2011) and Oser et al. (2012), this paper examines other salient forms of political participation. To avoid ruling out the effects on voting, the study will report the correlations in the exploratory section of the analysis, since voting activities should not be influenced by neoliberal conditions and as it carries a lower political impact relative to its cost.

Contemporary psychological models use education and socioeconomic status as primary variables for predicting individual political participation (Barrett and Brunton-Smith 2014; Cohen et al. 2001). Although political participation is higher among more educated citizens, who hold elite positions (Smets and van Ham 2013), the models imply that education and advancements in overall societal wealth should lead to higher political participation (Barrett and Brunton-Smith 2014; Cohen et al. 2001). However, statistical data do not back up this claim, as the political participation among US citizens remains low (Smets and van Ham 2013). The models do not address other underlying mechanisms, aiding to explain the low political participation of US citizens. While these models address the psychological

attitudes using variables such as self-esteem, locus of control, and collective, internal, external, and political efficiencies, they only explain those changes through education and socioeconomic status (Barrett and Brunton-Smith 2014; Cohen et al. 2001).

The psychological models build their predictions on the resource theory of political participation (Ojeda et al. 2020). People who possess more time, money, and civic skills are more politically active (Wang et al. 2017). Time is required for participating in offline politically related activities, money for making donations, and communicational and organisation skills for effective action (Wang et al. 2017). Status and education variables are good predictors for better communication skills, monetary resources, political knowledge, social integration with the elites, and higher self-esteem (Barrett and Brunton-Smith 2014).

However, this theory has its gaps, as it only looks at politically related acts but does not contrast them with other everyday activities (Ojeda et al. 2020). Additionally, the models do not look beyond the individual appraisals, neglecting environmental forces (Cohen et al. 2001) such as psychological motivations, social interactions, and the influence of salient social networks (McClurg 2003; Schlozman et al. 2018). The resource theory of political participation answers the questions regarding citizens' objective abilities. However, it neglects the realities surrounding their psychological and sociological motivations (Schlozman et al. 2018).

Thus, with the increasing demands from the labour market, making 60% of working Americans chronically stressed (Boyd 2021), this part of the paper examines the influence of subjective work-related stress on an individual's political participatory behaviours as a potential explanation for the decline in the psychological motivation.

2.1.3. Psychological Theories Explaining the Impacts of Work-Related Stress on Political Participation

Two independent sociological variables do not explain the psychological underpinnings of political participation as the resource theory of political participation proposes (Brady et al. 1995) since they account for 37% of the variance (Cohen et al. 2001). This paper uses the cognitive load theory and self-determination theory to explain an individual's cognitive overload and diminishing intrinsic motivation to describe the cognitive impacts of work-related stress on political participation.

**Cognitive Load Theory.** Cognitive load theory explains how personal cognitive capacities diminish when the subject is exposed to chronic environmental demands (Wosnitza et al. 2009). The cognitive load is associated with impaired working memory and comes from three sources: intrinsic, extraneous, and germane. Intrinsic load refers to the inherent difficulty of the task, extraneous load relates to the distracting elements in the environment, and germane cognitive load explains the integration process of novel information with existing knowledge (Wosnitza et al. 2009).

Environmental demands and uncertainties encouraged by integrating novel information create cognitive overload in the working memory, leading to cognitive depletion, pushing subjects to use system one thinking (Young et al. 2014). When these findings are contextualised with current working conditions, the stress from the uncertainties or opportunities leads to extensive cognitive overload (Fu et al. 2020).

With the increased number of novel elements in the environment, the contemporary workforce's intrinsic load is elevated, as working memory can simultaneously process only 3 to 7 elements (Wosnitza et al. 2009). Moreover, external distractions—extraneous load—further deplete one's working memory (Wosnitza et al. 2009). When combined, they increase brain fog (Balamurugan et al. 2020), diminishing the integration processes—germane load—with long-term memory, which does not further elevate the cognitive resources to attain desirable solutions on how to circumvent the stressors (Wosnitza et al. 2009), and hence decreasing the overall motivation through feelings of anxiety and apathy (Ryan 2019).

**Self-Determination Theory.** To understand the individual motivations for any action, self-determination theory (SDT) proposes psychological benchmarks through which indi-

viduals appraise their decisions (Deci et al. 2017). SDT is a macro-theory of an individual's subjective motivational drives.

Ryan and Deci (2000) have differentiated two types of individuals' motivations: intrinsic and extrinsic. Extrinsic motivation is governed by outside forces, predominantly through rewards and punishments. Contrastingly, intrinsic motivation denotes motivation desired by the agent (Ryan and Deci 2000). On the motivational continuum, two other motivations reflect the middle position (Wuttke 2020). The first is introjected motivation, through which a person is motivated to respond not to feel guilt or shame; even though they still dislike the external demands, they are pushed to act. Furthermore, identified motivation refers to self-selected behaviours regulated by ignored norms (Wuttke 2020).

As the primary need hypothesis states, individuals strive to feel autonomous—self-determined—in their actions (Deci et al. 2017). To initiate intrinsically motivated and satisfied behaviour, people have to feel autonomous, competent, and related to others. As the primary need hypothesis states, individuals strive to feel autonomous—self-determined—in their actions. The subject has to sense all three components, even though their salience can vary. For autonomy, one has to appraise feelings of internal will, not to experience an external control. The second component, competence, is felt when a person thinks they can influence the surrounding environment. Last, the relatedness aspect refers to the need to be connected to others. These aspects enable agents to strive for self-growth, as they are inherently intrinsically motivated (Deci et al. 2017).

Although intrinsic motivation alone is not predictive of political participation (Wuttke 2020), when combined with the cognitive load theory, it explains which activities people will allocate their time to (de Araujo Guerra Grangeia et al. 2016) since cognitive overload moderates intrinsic motivations (Schroeder and Adesope 2014). Therefore, as the resource theory of political participation states, time and money are finite resources (Ojeda et al. 2020), but so too is the working memory (Wosnitza et al. 2009). Hence, the allocation of those resources will predominantly go towards the three personal basic needs. And more personality overreaching behaviours, also needing intrinsic motivation, such as political participation, will be sidelined when demands from the environment are high (Vansteenkiste et al. 2020).

2.1.4. Work-Related Stress Appraisal of Two Types of Entrepreneurs

Work-related stress affects most workers across educational and socioeconomic circles due to increasing work demands (Boyd 2021), influencing intrinsic motivation via cognitive overload (Schroeder and Adesope 2014). Thereafter, to demonstrate why people across the socioeconomic spectrum do not participate in politics (Pew Research Center 2018), this section will explore the different appraisals of work-related stress for both socioeconomic ends of the workforce.

Axel Honneth (2019) divided the contemporary workforce into two camps; those who are highly qualified and can enjoy the marketised forms of securities and those who cannot. Although most of the workforce experiences work-related stress (Boyd 2021), the two groups appraise stress differently because of their positions in American society (Somers and Casal 2020). In this paper, the spectrum of job-market actors is divided into two groups; highly successful workers are labelled as always-improving entrepreneurs, and those who are least successful are defined as failed entrepreneurs.

Lazarus and Folkman (1984) proposed that work-related stress emerges due to an individual's two-step appraisal process when a stressful state is encountered. Hence, a stress response is triggered when a situation is perceived as a threat—primary appraisal—and if the individual does not have sufficient resources to cope with the stressor—secondary appraisal (Lazarus and Folkman 1984). When a stressor is appraised as a threat—harm is anticipated, producing emotions such as fear or anxiety (Mühlhaus and Bouwmeester 2016).

There are two types of individual stress coping mechanisms: problem-focused and emotion-focused coping (Landy and Conte 2012). The first refers to strategies for eliminating the stressor, and the latter strives to regulate emotional reactions. However, nei-

ther mechanism can overcome chronic stress (Landy and Conte 2012), making chronic work stressors of particular interest because they might create persistent barriers towards political participation as stress affects one's motivation, cognition, and future behaviour (Ojeda et al. 2020).

**Stress Appraisal of the Always-Improving Entrepreneur.** When using the empirically well-established Karasek's demand–control model, the always-improving entrepreneur has a job scoring high on both psychological demands and perceived control (Van der Doef and Maes 1999). This formula produces a highly engaged labourer whose conduct is in line with the injunctive work norms proposed by the meritocratic ideal (Jacobson et al. 2020). This constellation leads to identified and intrinsic motivational behaviours. However, the subject's abilities are contested by the ever-changing labour environment, stressing them to refine their abilities as the person–environment fit model proposes (Caplan and Harrison 1993). Hence, most always-improving entrepreneurs are driven by identified motivational behaviours subjugated to normative behaviours incapable of challenging the status quo.

The chronic struggle to deliver unlimited, poorly defined, dynamic demands from the labour market environment occupied by other exceedingly skilled counterparts results in Type A behaviour (Petticrew et al. 2012). People with Type A behaviour are ambitious, impatient, hostile, and urgent—they are in a constant endeavour to achieve more in less time (Landy and Conte 2012), suggesting higher loads on their cognition. This behavioural pattern has manifested itself in the phenomenon called the fear of missing out (FOMO), anxiety over being absent from a possible rewarding experience, always in comparison with others (Jood 2017). Around 62% of higher status workers have this fear, which negatively correlates with their life satisfaction (Jood 2017), suggesting identified motivational patterns.

To avoid losing their high-status position, which enables them to obtain their intrinsic needs, always-improving entrepreneurs rely on the constant comparison with the dominant ideological ideal (Meurs et al. 2010). The self-categorization model of stress combines Lazarus and Folkman's (1984) stress model with the self-categorization theory. When an individual's social identity is cognitively activated, the evaluation of the stressor is mediated by the group's membership. Moreover, the group's norms, perceived resources, and social support will play a role in the secondary appraisal. Hence, for highly skilled workers to maintain their labour position, they have to appraise their stressors as an opportunity for growth, focusing on problem-focused coping (Meurs et al. 2010).

The constant comparison with others, always having to change to fit the demands of the labour market, is resulting in a self-regulatory crisis (Jood 2017). It further induces a stress response, as there is a discrepancy between what is desired and what is occurring in reality (Meurs et al. 2010) since the behaviours are not intrinsically motivated. Overall, the always-improving entrepreneurs will focus their basic intrinsic needs towards managing their high-status position instead of collectively engaging in political actions.

**Stress Appraisal of the Failed Entrepreneur.** The second type is the failed entrepreneur. When illustrated via the demand–control model, these workers score high on psychological demands and low on perceived control (Van der Doef and Maes 1999). Their contracts are alternative without the stable securities that the always-improving entrepreneur enjoys (Katz and Krueger 2019). Unstable securities create a higher cognitive overload since they have many more stressors to take care of with limited resources (Matthews et al. 2019).

Failed entrepreneurs score low in all three aspects proposed by SDT (Deci et al. 2017), as only 33% of all Americans are engaged in their work, and 18% of employees believe that they are fairly evaluated (Gallup 2021). Hence, the aspiring and supporting aspects are nonexistent in the precarious workplace (Deci et al. 2017). Furthermore, failed entrepreneurs experience shame since their careers are not valued by society due to the low perceived status of their work (Frost 2016), reflecting their motivations to work either by external motivation induced by monetary reward or by the introjected motivation to alleviate feelings of shame.

Moreover, failed entrepreneurs are easily replaceable and have no autonomous control over their work lives (Van der Doef and Maes 1999). The lack of control comes from the overall uncertainty of the work, working conditions, and economic vulnerability (Allan et al. 2021) Person–environment fit is also low (Caplan and Harrison 1993) because of the subjectively appraised insecurity, instability, and powerlessness (Creed et al. 2020).

These precarious working conditions and subjective attitudes towards one's work disrupt one's financial capabilities and psychological purpose derived from work, lowering extrinsic and intrinsic motivations (Deci et al. 2017). The inability to change one's working conditions leads to learned helplessness (Nicolas et al. 2016), pushing failed entrepreneurs to use emotion-focused coping, and creating an environment where stressors become chronic (Landy and Conte 2012). Hence, chronic stress establishes a feedback loop of inaction (Meurs et al. 2010) and alienates the subject from society (Shantz et al. 2015), potentially explaining why people with lower economic status do not participate in politics (Emmenegger et al. 2015).

### 2.1.5. Concluding Remarks for the First Hypothesis

The first hypothesis of the paper argues that persistent stress-inducing neoliberal working conditions across all socioeconomic sectors (Ojeda et al. 2020) lead to depletion of one's cognitive resources, which raises the barriers to effectively allocating those resources towards higher intrinsic needs such as political participation (Creed et al. 2020; Gallup 2021), affecting the overall psychological motivation to participate in politics. The hypothesis builds upon research by Ojeda et al. (2020). The Ojeda et al. (2020) study utilised the resource theory model to examine the correlation between overall stress and voter turnout among the US eligible voters, using the GSS dataset. This paper aims to take this proposition one step further. Hence, the first hypothesis: *work-related stress will be negatively related to political participation*.

### 2.2. Offline Social Leisure and Political Participation

### 2.2.1. Leisure—Critical Theory

The contribution of the Frankfurt school to furthering the impacts of leisure in political discourse lies in the idea that people's confirmatory behaviours are also promoted in their post-work leisure time via the entertainment industry and marketised products, advancing the capitalist ideal (Zambrana 2013). Their claims are interesting for contemporary America since the global market offers various institutionalised coping instruments such as self-help books, celebrity tabloids, exercise programs, or meditation apps to lower people's stress—induced by working conditions—and promote hope (Hefty 2020).

Adorno (2016) saw leisure commodities as another avenue for capitalising on people's stress coping mechanisms by engaging them in meaningless activities. Those activities alienate US citizens and suppress their critical thinking. For him, American entertainment fulfils this purpose. It gives American citizens promises of success, mimicking their desires in fictional worlds (Adorno 2016). Alternatively, in popular talk shows, where idolised celebrities and entrepreneurs are frequently asked about their hardships before their successes to exemplify deliberate work as the tool for overcoming the structural constraints of the system (Driessens 2013) since these social roles are the only rescue positions from an otherwise deep-rooted position of social subordination (Littler 2013). Ultimately, the passive nature of the new form of leisure strips down one's agency since it does not require the spectator to engage in any other activity except consumption (Marcuse 1970).

Debord (2002) furthered the claims of the Frankfurt school as he described how capitalistic ideology creates normative behaviours. Spectacles—physical and intellectual goods—are produced in capitalism, and they only refer to the capitalistic ideal, producing unified discourse from which citizens cannot untie themselves. Those living in a capitalistic society accumulate—consume—the created spectacles, as they appear to be the only viable option. Thus, they do not revolt against the overall proposed options of the ideology (Debord 2002).

The claims by critical theorists suggest that with the rise of the entertainment industry, leisure time was dramatically transformed by the television, which enabled mass consumption of ideological products, further perpetuating passive forms of leisure (Adorno 2016). These claims are products of their observations (Habermas 1988). Therefore, it is necessary to contextualise their claims with recent data to show the importance of leisure time in the political participation discourse.

### 2.2.2. Empirical Evidence for the Lack of Social Leisure

Throughout the second half of the 20th century, technological growth and the decline of social capital have been unprecedented (Antoci et al. 2013; Dalton and Klingemann 2011). Putnam (2020) demonstrates that technological progress has been the key driver for the erosion of social connectedness because leisure has become increasingly individualised as people have not needed to coordinate their tastes with those of others. His claims were experimentally tested, and the marketisation of civil life has indeed led to the diffusion of values, enabling people to circumvent social relations (Bartolini et al. 2011).

The data on leisure activity preference among American full-time workers suggest that more engaged workers, described in this paper as always-improving entrepreneurs, prefer active coping forms of leisure (Somers and Casal 2020) to stay focused (Trenberth and Dewe 2002), in the competitive landscape (Lee and Lee 2015), and not to experience FOMO (Jood 2017). Those findings were confirmed by Lee and Lee (2015), who found that workers utilise their leisure time through entertainment, knowledge acquisition, and subsequently around relaxation and physical exercise.

For failed entrepreneurs, the coping strategy does not resemble problem-focused coping but emotional-focused coping, since the primary function of their leisure time is to reframe or distract themselves from the stressor (Trenberth and Dewe 2002). This coping style corresponds to leisure choices associated with entertainment, relaxation (Lee and Lee 2015), and intoxication (Somers and Casal 2020).

Yet, for both groups, hobbies and social leisure are rarely mentioned (Lee and Lee 2015).

As the Frankfurt School proposed, a contemporary employee's leisure time has become a solitary endeavour (Adorno 2016). There are two distinct leisure coping styles for workers (Trenberth and Dewe 2002). Both active, challenging and passive, corrective styles are in line with the problem-focused coping and the emotional-focused coping division following the neoliberal ideal (Trenberth and Dewe 2002). Thus, leisure time becomes either an extension of one's work or a passive consumption endeavour (Elliott 2018), stressing the importance of studying this variable to identify other factors overlooked by the resource theory of political participation.

### 2.2.3. The Entrepreneurial In-Groups and Their Impacts on Collective Identity

As workers spend their leisure time solitarily (Somers and Casal 2020), and since work has become the primary source of merit and the means for pursuing one's liberation (Carter 2006), both entrepreneurial selves have to strip their traditions and communal identities to rise in the ever-changing opportunity-rich world (Cushman 1990). Those tendencies are expressed in statistics, as 61% of Americans report feelings of loneliness (CIGNA 2020; Sønderby and Wagoner 2013). Moreover, the work relationships do not fill in the gap, as 80% of Americans do not have close ties with their colleagues (Gallup 2021). Therefore, the modern self loses its social relationship because of work demands and leisure choices.

The diminishment of social leisure and the rising of loneliness rates imply that both entrepreneurial selves are not developing close relationships with their in-groups. Furthermore, this development is problematic since in-group conversations foster discussions around societal questions (Mair 2002) and alleviate personal resource constraints around political participatory behaviours (McClurg 2003). Moreover, the lack of close relationships diminishes collective identity cultivated through private in-group relationships that offer kinship and promote trust, obligation, and reciprocity (Feldmeyer et al. 2017). To further examine this development, social identity theory will be applied.

**Social Identity Theory.** Social identity theory operates on the presumption that individuals want to maintain and enhance their self-concept regarding their social identity by comparing themselves with other groups based on a given set of values (Peterson and Stewart 2020; Tajfel and Turner 2004). Through this cognitive process, individuals subconsciously develop group identities based on similarities and perceived differences. To simplify the social environment, they categorise others as part of an in-group, leading to favouritism. Contrastingly, when others do not resemble their social identity, they categorise them as part of an out-group. When group membership is salient, individual members act under the in-group norms. Nevertheless, those abstract norms do not disqualify the individual distinctiveness from emerging (Tajfel and Turner 2004).

When looking at labourers in neoliberal society, always-improving entrepreneurs are viewed as the favourable in-group and the failed entrepreneurs as out-group members of the society. To be part of the in-group, individuals have to identify with the injunctive norms related to the neoliberalised meritocratic ideal (Reed et al. 2007). Furthermore, to maintain a positive self-concept and be optimally distinct in an uncertain competitive environment, individuals have to strive for high-status positions, as those positions carry less uncertainty (Peterson and Stewart 2020). Thus, always-improving entrepreneurs, the neoliberal in-group, operate with the meritocratic ideal through which they are optimally distinct (Littler 2013).

**Always-Improving Entrepreneurs.** The always-improving entrepreneur's self-concept is linked to an in-group of successful workers (Peterson and Stewart 2020). They aspire to move up the social ladder to acquire high-status positions (Vignoles et al. 2006). These ideals are explained by the motivated identity construction theory (MICT), advancing the notions of social identity theory (Vignoles et al. 2006). To be authentic and to have higher self-esteem in the ever-changing complexity rich environment, one rejects belonging to the specific groups in favour of being optimally distinct (Peterson and Stewart 2020). Moreover, one attaches to those who positively appraise personal complexity to satisfy belonging and uniqueness (Peterson and Stewart 2020). Last, to signal personal success, one has to devalue the impact of social structures (Guilbaud 2018), resulting in a downward comparison and lower pro-social behaviour (Iacoviello and Lorenzi-Cioldi 2019). Hence, the dominant in-group disvalues the prolonged development of close relationships because it impairs individual uniqueness and results in trade-offs with career-life time, through which they construct their social identities (Iacoviello and Lorenzi-Cioldi 2019).

**Failed Entrepreneurs.** Meanwhile, failed entrepreneurs are perceived as the inferior out-group (Standing 2021). Members of the failed entrepreneurial group try to escape their group through conspicuous goods signalling, as their social mobility is impaired when the out-group membership becomes salient by their occupational status that increasingly offers less options for social mobility (Sun et al. 2020). Failed entrepreneurs also use the domain disengagement strategy (Berjot and Gillet 2011); they make fewer contributions towards their own in-group because they want to decrease the influence of the stigmatised in-group identity on themselves. Hence, they mimic the always-improving entrepreneurs to obtain favourable comparisons with the dominant in-group (Berjot and Gillet 2011; Outten et al. 2008).

As McClendon (2020) empirically described (, failed entrepreneurs engage in upward comparison, resulting in admiration or envy towards more successful members of society (van de Ven 2015). Thereafter, they are more likely to support policies against their own in-group because of status motivations to increase their relative positions within the deprived group (Duménil and Lévy 2013); losing their close relationships by devaluing one's in-group, as this becomes the tool for moving up through the social hierarchy (Iacoviello and Lorenzi-Cioldi 2019).

2.2.4. Participatory Efficacy as a Predictor for Collective Action

Neoliberal injunctive norms associated with the meritocratic ideal (Reed et al. 2007) and the individualised leisure choices (Bartolini et al. 2011) have led Americans to a

persistent feeling of loneliness because of nonexisting close relationships (Sønderby and Wagoner 2013), diminishing the notions of collective identity (Stebbins 2016). The lack of close social interactions can be another contributor to low political participation because all components of the individual-level characteristics of political action have a social component to them (Campbell 2013).

An ineffective and individualised form of political participation is well-described by political psychologist Hersh (2020), who has named the effect political hobbyism. Even though Americans are more politically informed, they do not act to accelerate policy change effectively, but use politics to be emotionally fulfilled via the entertainment streams or to appear more knowledgeable among their peers (Hersh 2020). Those motivations closely resemble the leisure activities preferred by both entrepreneurial groups—knowledge acquirement and entertainment. Thus, the individualised form of political engagement does not necessarily lead to political participation (Thomas and Louis 2013).

**Collective Identity and Collective Action.** To challenge or protect the political status quo, an action commencing from collective identity must be apparent (Stuart et al. 2018). van Zomeren et al. (2004) have experimentally shown and explained the components for collective action via the introduced dual-process model. For individuals to be collectively active, they have to know whether others share their opinions—emotional social support— but also whether the potential movement has enough resources to stay united under distress—instrumental social support. Both those factors are necessary for the individual to be motivated to partake in collective action, since one uses those components for their cost–benefit analysis (van Zomeren et al. 2004).

The social identity model of collective action furthers the dual-process model as it examines the main predictor for collective action, one's social identity (van Zomeren et al. 2008). The model operationalises the identity variable based on two variables: cognitive centrality—the salience of group membership in self-concept—and affective in-group ties— the sense of connection with other group members (Cameron 2004). Hence, the closeness of one's relationship and the inclusion of the other in the self-concept (Tausch et al. 2011) should be predictive of collective action.

The perceived collective identity is established through the group's offline emotional social support, which further cultivates the notions of instrumental social support, positively influencing personal and collective efficiency beliefs (van Zomeren et al. 2008). Collective identity is especially fostered through offline strong social ties offering kinship, promoting trust and obligation, and creating reciprocity (Feldmeyer et al. 2017). Ongoing reciprocity between both actors in a close relationship makes it easier to instigate mobilisation (Williams 2007), making the diminishment of American offline social leisure a concerning factor for political participation rates.

The advancement of collective identity via offline social interactions has been empirically tested by Williams (2007), who found that online social networks expand social capital, but those newly acquired relationships are not solid friendships with strong emotional support, as the low entry and exit costs created by the platform's environment establishes weak-tie relationships. Those findings are translated into collective action as the latter aspect of the dual-process model—instrumental social support—cannot be inferred on social media (Kende et al. 2015), explaining why activism on social media platforms is not predictive towards offline collective action as it only magnifies the influence of those who are already politically active, but does not lead to the creation of new social ties (Oser et al. 2012).

**Collective Identity and Participatory Efficacy.** Although collective identity is strongly associated with collective action (Cameron 2004), there is a hidden contradiction between the dual-process model and the social identity model of collective action (van Zomeren et al. 2012). In those models, the individual who holds a strong identity and believes in instrumental social support should not be engaged in protesting because they already believe in the group's success (van Zomeren et al. 2012). Therefore, a newly identified variable in political psychology comes into play. Participatory efficacy comes second after

collective identity in predicting political participation, and it explains the action paradox apparent from the dual-process model (Bamberg et al. 2015).

A recent study looked at the bridging correlation between participatory efficacy and collective identity (Bamberg et al. 2015). Participatory efficacy beliefs—that one can effect social change through one's actions—served as a predictive bridge between the group and individual beliefs about potential collective action (van Zomeren et al. 2012). Bamberg et al. (2015) discovered that participatory efficacy explains the cost–benefit calculation toward collective action, as collective contribution will only occur when participatory efficacy is strong.

Hence, one must look at the individual's participatory efficacy perceptions of their collective abilities to predict collective action reliably (Thaker et al. 2018), retaining a subjective cost–benefit analysis (Thomas and Louis 2013), since perceived efficacy plays a crucial role in directing one's behaviour for political participation (Bandura 2000). This variable can be diminishing in the American population who spend less time in offline social leisure activities, thus potentially affecting the low political participation numbers.

2.2.5. Concluding Remarks for the Second and Third Hypotheses

To tackle the second neglected part in the resource theory of political participation —the social motivation—the diminishing time spent on offline social activities needs to be examined, as offline social networks play a substantial role in motivating individuals to be politically active (Klandermans and van Stekelenburg 2013; Schlozman et al. 2018). Therefore, the second and third hypotheses advance the resource theory of political participation as they bring new environmental variables into the equation.

Although work/leisure produces different identity appraisals for both entrepreneurial groups, suggested by the social identity theory (Iacoviello and Lorenzi-Cioldi 2019), both entrepreneurial camps have less-close relationships (Sønderby and Wagoner 2013), and they participate less in social leisure activities (Somers and Casal 2020), as the critical theorist observed (Adorno 2016). Notably, with the rise of social media connectivity, close relationships are not supported through those platforms (Williams 2007), suggesting offline social leisure is the avenue where people nurture their close ties. Ultimately, those connections translate to the higher notions of collective identity necessary for political participation (van Zomeren et al. 2008).

Since collective identity and individual cooperation towards the group are fostered through offline social leisure, it should also spill over to participatory efficacy as group-based hobbies and close friendships encourage group mentality (Thomas et al. 2017). Participatory efficacy is the bridging variable of collective identity induced by offline social leisure and the action potential resulting in political participation (Bamberg et al. 2015).

It has been confirmed that being a member of voluntary, community-based initiatives or religious groups is predictive of political participation (Lim 2008; Lim and MacGregor 2012). However, there have not been studies with representative samples focusing on group hobbies or offline conversations with close friends (Lim 2008; Teney and Hanquinet 2012).

Hence, the second and third hypotheses: *offline social leisure will be positively related to political participation*, and *participatory efficacy will positively mediate the association between offline social leisure and political participation*.

**3. Current Study**

The quantitative correlational study will answer two research questions using the data from the General Social Survey to explore relationships of the independent variables, work-related stress and offline social leisure, with the dependent variable denoting political participation. A mediating variable, participatory efficacy, is used to explain further the relationship of offline social leisure on political participation.

The hypotheses are (see Figure 1): (H1) Work-related stress will be negatively related to political participation; (H2) offline social leisure will be positively related to political participation; (H3) participatory efficacy will mediate the association between offline social

leisure and political participation. Education and socioeconomic status were controlled for all variables, ensuring that those findings are universal for both ends of the US workforce.

For explanatory purposes, the paper aims to rule out the influence of the independent variables on voting. Furthermore, the study will evaluate the model fit of the proposed work/leisure pathways with the control variables of education and socioeconomic index, which are being used as primary variables for examining political participation. Last, it will also examine the relationship between work-related stress and offline social leisure to determine whether there is a direct relationship between those two variables.

To increase the internal validity of the results, the mediation regression analysis will be conducted for the significant pathways of the independent variables declared by structural equation modelling. Furthermore, to crosscheck the external validity of the model's significant pathways, an additional structural equation modelling analysis will be run on the 2014 European Social Survey sample.

## 4. Methods

### 4.1. Dataset and Sample

The study used the General Social Survey (GSS) 2014 dataset to test the hypothesised model. The dataset was created by the National Opinion Research Center at the University of Chicago (National Opinion Research Center (NORC) 2019). The strength of the GSS lies in its sample representativeness. It allows for more valid conclusions since stress research relies on city-specific samples (Ojeda et al. 2020). The repeated cross-sectional survey uses full-probability sampling of noninstitutionalised US adults aged 18 and over, and it is gathering its longitudinal data for 49 years based on voluntary participation (Kim et al. 2017). After the US consensus, the GSS is the most used data source in the social sciences, as 25000 papers have used this sample in their research (NORC 2019).

The study used the GSS 2014 survey ballots, as the 2014 survey was the most recent year when the questions around participatory efficacy, political participation, and one component of offline social leisure variable—belonging to a social, sporting, or cultural group—were recorded (NORC n.d.a). Another time when all those questions were raised was in 2004. The study could not utilise the total sample size due to the nature of the survey. Some attitudinal questions had been asked only to a handful of participants to ensure the project's feasibility, as the 2014 questionnaire consisted of 926 items. Hence, the respondents were divided into three ballot groups, still ensuring the representatives of the sample. The study had to omit ballot C as questions surrounding the second component of the offline social leisure variable were only asked in ballots A and B (NORC n.d.a).

The preliminary sample comprised 384 participants from ballots A and B. Participants who were not eligible to vote in the US ($n$ = 33) and those who poorly comprehended the questions ($n$ = 56) were excluded from the study. Last, the interquartile range method was used to identify two outliers.

The final sample consisted of 295 participants ($M_{age}$ = 44.49, $SD$ = 13.43, range: 21–77). The majority of the sample were women (55.9%). On average, the sample's highest attained year of education was 14 years ($M_{education}$ = 14.14 $SD$ = 2.61, range: 6–20), and the average participant scored 48 on the socio-economic index ($M_{status}$ = 48.34, $SD$ = 23.00, range: 13–93).

### 4.2. Measures

#### 4.2.1. Work-Related Stress

The first independent variable used a question regarding respondents' workplace self-reported stress attitudes (1 = *always* to 5 = *never*). The respondent was asked the following question: "how often do you find your work stressful?" (NORC n.d.b). Self-reported answers regarding singular life events are standardly used as a tool to assess stress after physiological traits and experimental stress primes (Ojeda et al. 2020).

### 4.2.2. Offline Social Leisure

The second independent variable consists of two items and represents a latent variable denoting offline social leisure. The first item asked respondents to indicate whether they belong to a social, sporting, or cultural group (1 = *belongs and active* to 4 = *never belonged*). The second item was created from three underlying variables—sharing one commonality—the amount of time focused on close social interactions. The respondents were asked how often they spend a social evening with relatives, neighbours, and friends (1 = *almost daily* to 7 = *never*). The three variables were recoded (1 to 4 scale) and computed to carry equal weight to the first item. The items were picked as they represent respondents' allocation of their leisure time towards social activities. These items were accounted for in the Somers and Casal study (2020) when computing for social coping leisure activities as they both foster close relationships among the participants (Roberts 2018).

### 4.2.3. Participatory Efficacy

One item is in the mediation variable. The respondents were asked to indicate how important it is for them to be active in a social or political group (1 = *not at all important* to 7 = *very important*). The scores were reversed coded from the original variable scores. Thus, the lower scores represent higher individual importance. The question resembles the question used in Bamberg et al. (2015) study that looked at participatory efficacy. The researchers used this question to find the indicator for the latent variable, "How strong is your intention to actively and regularly participate in a local TT group?" (Bamberg et al. 2015).

### 4.2.4. Political Participation

It is common in political research to sum multiple components of political participation to represent the latent variable (Oser et al. 2012; Quintelier and van Deth 2014; Strömbäck et al. 2017; Weinschenk and Panagopoulos 2014). The dependent variable represents a latent variable and has six items. All six items are used repeatedly by the Pew Research Center (2009) to measure political participation. The respondents were asked to indicate their behavioural tendencies regarding each politically participatory activity (1 = *have done it in the past year* to 4 = *have not done it and would never do it*), "Signed a petition," "Took part in a demonstration," "Attended a political meeting or rally," "Contacted politician or civil servant to express view," "Donated money or raised funds for social or political activity," "Contacted media to express view" (NORC n.d.c).

### 4.2.5. Voting

The second dependent variable, used for the exploratory analysis, corresponds to voting behaviour; the respondents were asked whether they had voted in the last presidential election (1 = *yes* to 2 = *no*). The last presidential election for this sample took place in 2012.

### *4.3. Data Analysis Strategy*

The study's hypotheses and analysis plan were preregistered at the Center for Open Science. The raw dataset, modified dataset, SPSS syntax with all the explained steps, and the SEM analysis file can be found through the link: https://osf.io/9cpx2/ (accessed on 23 May 2021). Moreover, to increase the internal validity of the results, the mediation regression analysis was conducted for the significant pathways of the independent variables declared by the structural equation modelling. Those results can be viewed in Appendix B. Furthermore, to crosscheck the external validity of the model's significant pathways, an additional structural equation modelling analysis was run on the 2014 European Social Survey sample containing 27,604 respondents. Its methodology and result sections can be viewed in Appendix C.

### 4.3.1. Principal Component Analysis

Since the analysis works with two latent variables, offline social leisure and political participation, principal component analysis (PCA) with the Varimax rotation has been used

to determine how much and whether all those underlying variables adequately explain the total variance of the latent variables. The analysis only used those components that score above one on the eigenvalue (Hair et al. 2018).

The internal reliability of the underlying components is further tested by Cronbach's Alpha and the mean interitem correlations. The mean of interitem correlations is used as an additional measure, as Cronbach's Alpha alone can be biased because of the number of measured components (DeVellis 2021).

For the proposed latent variable, the acceptable minimum for the total variance explained by the underlying variables is 50% (Field 2018), with factor loading resembling correlation coefficients between the variables and the latent factor, and commonalities representing correlations to be both at least 0.03 (Field 2018). Moreover, the latent variables need to pass the reliability checks. The acceptable mean interitem correlation ranges from 0.15 to 0.50 (Bañales et al. 2020). For Cronbach's Alpha, the acceptable value has to be above 0.70 (Cortina 1993).

### 4.3.2. Structural Equation Modelling

For the main analysis, structural equation modelling (SEM) via SPSS Amos 25 was used to estimate the effects on the hypothesised paths. The analysis had not been planned to test alternative models, and the items were treated as continuous since all were measured at least with a Likert scale (Muthen 2021) except the voting variable.

The goodness of fit is used to assess whether the hypothesised constructs explain the dataset (Nunkoo and Ramkissoon 2012). Relative chi-square is favoured over chi-square goodness of fit since the test is less sensitive to sample size. The test should not exceed values above 2 to establish whether there are differences between expected and observed values proposed by the model (Barbara 2021). Furthermore, the root mean square error (RMSEA), standardised root mean square residual (SRMR), comparative fit index (CFI), and the Tucker–Lewis Index (TLI) weigh the best possible model with the worst possible model, comprising all variables independent of each other (Peugh and Feldon 2020). The model fit is deemed very good when values are greater than 0.95 for CFI and TIL and below 0.05 for RMSEA and SRMR (Bañales et al. 2020).

To assess all effects, the model paths are separated into direct, indirect, and total effects. To measure the indirect and total effects and the 95% confidence intervals (CI) for all paths, the bootstrapping procedure of 10,000 samples through an AMOS user-defined estimand was used (Amos Development Corporation 2010). The 95% CI is deemed significant if neither the lower nor upper bound include zero values (Kline 2015).

### 4.4. Covariates

To ensure the universality of the findings on both ends of the US workforce and explain the current psychological model of political participation—the resource theory of political participation (Barrett and Brunton-Smith 2014)—education and socioeconomic variables were used as control variables. The highest year of school completed was used as a proxy for education, ranging from 0 to 20 years. The interviewer had evaluated the social–economic status of the participants via the 2010 socioeconomic index. The 2010 socioeconomic index maps 539 occupations recognised by the 2010 Standard Occupational Classification and assigns prestige ratings, from 0 to 100, to the occupations based on total income, hourly wage, and educational credentials (NORC 2015).

### 5. Results

Table 1 illustrates the descriptive statistics for all primary variables used in the analysis. All variables had normal univariate distributions.

**Table 1.** Descriptive Statistics (*n* = 295).

| Variable | *M* | *SD* | *SE* | Skewness | Kurtosis |
|---|---|---|---|---|---|
| Signed a petition | 2.09 | 0.97 | 0.06 | 0.46 | −0.80 |
| Took part in the demonstration | 3.14 | 0.77 | 0.05 | −0.60 | −0.08 |
| Attended political meeting or rally | 2.93 | 0.88 | 0.05 | −0.50 | −0.46 |
| Contacted politician to express view | 2.72 | 1.07 | 0.06 | −0.37 | −1.10 |
| Donated money or raised funds | 2.60 | 1.10 | 0.06 | −0.20 | −1.27 |
| Contacted media to express view | 3.31 | 0.77 | 0.05 | −1.05 | 0.88 |
| Voted in 2012 election | 1.29 | 0.46 | 0.03 | 0.92 | −1.16 |
| Work-related stress | 2.94 | 1.02 | 0.06 | −0.02 | −0.22 |
| Social hobbies | 2.80 | 1.17 | 0.07 | −0.49 | −0.26 |
| Social evenings [a] | 3.14 | 0.91 | 0.05 | −0.06 | −0.61 |
| Participatory efficacy (R) | 3.34 | 1.54 | 0.09 | 0.51 | −0.02 |
| Highest year of school completed | 14.14 | 2.61 | 0.15 | 0.19 | −0.17 |
| Respondent's socioeconomic index | 48.34 | 23.00 | 1.34 | 0.25 | −1.18 |

*Note.* The reverse coded variable is denoted with an (R). [a] This variable was computed from three underlying variables a social evening with relatives, neighbours, and friends. The three variables were recoded (1 to 4 scale).

## 5.1. Principal Component Analysis

For the first latent variable, political participation, the PCA looked at six underlying variables. The Kaiser–Meyer–Olkin measure was 0.83, with all individual measures greater than 0.78. Bartlett's test of sphericity was statistically significant ($p < 0.001$). Furthermore, an inspection of the correlation matrix indicated that all six variables were at least correlated above 0.30. The analysis proposed one component structure with an explained variance of 50.24%. The varimax orthogonal rotation exhibited that the adequate simple structure should retain all six variables since the average factor loading score was 0.70. All six-component loadings and communalities are visible in Table 2.

**Table 2.** Rooted Structure Matrix for PCA with Varimax Rotation of Both Latent Variables.

| Variable | Rotated Component Coefficient | |
|---|---|---|
| | Component 1 | Communalities |
| *Political Participation Latent Variable* | | |
| Attended political meeting or rally | 0.821 | 0.675 |
| Contacted politician to express view | 0.787 | 0.619 |
| Took part in demonstration | 0.715 | 0.511 |
| Contacted media to express view | 0.703 | 0.494 |
| Donated money or raised funds | 0.645 | 0.416 |
| Signed a petition | 0.547 | 0.299 |
| *Offline Social Leisure Latent Variable* | | |
| Social hobbies | 0.764 | 0.584 |
| Social evenings | 0.764 | 0.584 |

To further ensure the internal reliability of the proposed variable, Cronbach's alpha was 0.79, and the mean interitem correlations indicated a score of 0.40. Hence, the proposed latent variable could be further used in the analysis as it reliably represents 50.24% of the variance of all proposed politically participatory activities.

For the second latent variable, offline social leisure, the PCA analysis looked at two underlying variables. The Kaiser–Meyer–Olkin measure was 0.50. Bartlett's test of sphericity was statistically significant ($p < 0.01$). Furthermore, an inspection of the correlation matrix indicated they were correlated with a score of 0.58. The analysis proposed one component structure with an explained variance of 58.43%. The varimax orthogonal rotation exhibited that the adequate simple structure should retain all two variables since the average factor loading score was 0.76. The two-component loadings and communalities are visible in Table 2.

To further warrant the internal reliability of the proposed variable, only the mean interitem correlations could have been employed because Cronbach's Alpha cannot be used for two items (Rammstedt and Beierlein 2014). The mean interitem correlation indicated a score of 0.17. Hence, the suggested latent variable can be further used in the analysis as it reliably represents 58.43% of the variance of all proposed offline social leisure activities.

*5.2. Bivariate Correlations*

First, the bivariate correlations found that work-related stress was positively and significantly associated with political participation ($r(293) = 0.15$, $p < 0.01$) and was not significantly associated with voting. Second, offline social leisure was positively and significantly associated with political participation ($r(293) = 0.43$, $p < 0.001$) and voting ($r(293) = 0.23$, $p < 0.001$). Third, offline social leisure was also positively and significantly related to participatory efficacy ($r(293) = 0.23$, $p < 0.001$). Fourth, participatory efficacy was positively and significantly associated with political participation ($r(293) = 0.37$, $p < 0.001$) and voting ($r(293) = 0.27$, $p < 0.001$).

*5.3. The Main Analysis*

To check for multicollinearity, the variance inflation factor indicated values below 1.78, and the tolerances were above 0.98. Therefore, all assumptions about influential points, multinormality, and multicollinearity were checked.

The SEM tested the hypothesised direct and mediating relations. For the model to still be overidentified, to gain degrees of freedom, the nonsignificant status control path for the participatory efficacy was deleted ($\beta = -0.04$, $p = 0.59$), This step is not seen as problematic since both control variables are highly correlated with each other ($\beta = 0.63$, $p < 0.001$). Furthermore, the nonsignificant educational control path estimating political participation was retained for the exploratory part of the analysis.

Model fit indicators showed that the hypothesised model paths were good fits for the data ($\chi^2/df = 3.40/3 = 1.13$; RMSEA = 0.02; CFI = 0.99; TLI = 0.99; SRMR = 0.02).

All direct paths are denoted by a standardised coefficient ($\beta$), representing the effect sizes. The main direct paths are stated in Figure 2. All other effects, such as unstandardised coefficients, confidence intervals, and standard errors, are in Appendix A.

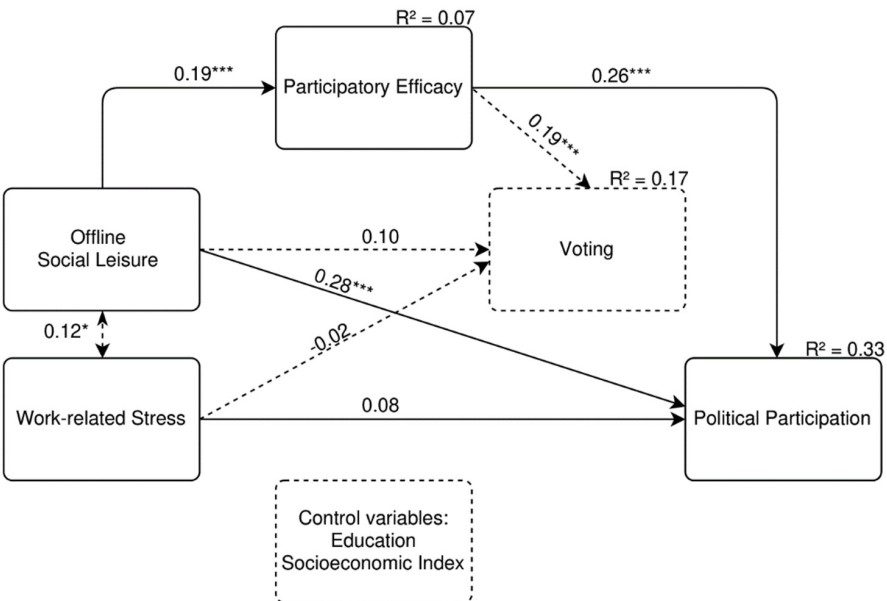

**Figure 2.** Pathways' effect sizes for the conceptual model of the relationship between work-related stress/offline social leisure and political participation. The political participation variable does not contain voting. The full lines denote the main analysis pathways, whereas dotted lines represent exploratory analysis pathways. * $p < 0.05$ *** $p < 0.001$.

### 5.3.1. Results for All Three Hypotheses

The first hypothesis proposed that work-related stress is negatively associated with political participation; the path analysis did not find a significant correlation between work-related stress and political participation ($\beta = 0.08$, $p = 0.09$).

The second hypothesis had expected to see a direct positive relationship between offline social leisure and political participation. For this path, the analysis found a significant positive relationship ($\beta = 0.28$, $p < 0.001$).

The third hypothesis a positive mediating effect between offline social leisure and participatory efficacy concerning political participation. The mediating effect is significant ($\beta = 0.05$, $p < 0.001$), with offline social leisure positively interacting with participatory efficacy ($\beta = 0.19$, $p < 0.001$), and participatory efficacy positively interacting with political participation ($\beta = 0.26$, $p < 0.001$). Moreover, the total effect of offline social leisure on political participation was also found to be significant ($\beta = 0.33$, $p < 0.001$).

### 5.3.2. Exploratory Results

First, work-related stress ($\beta = -0.02$, $p = 0.82$) and offline social leisure ($\beta = 0.10$, $p = 0.09$) are not correlated with voting. The mediating effect is significant ($\beta = 0.04$, $p < 0.01$), with offline social leisure positively interacting with participatory efficacy ($\beta = 0.19$, $p < 0.001$), and participatory efficacy positively interacting with political participation ($\beta = 0.19$, $p < 0.001$). Moreover, the total effect of offline social leisure on political participation was also found to be significant ($\beta = 0.14$, $p < 0.05$).

Second, the proposed model fitted the data well, as shown by the model–fit indicators. The model explained 33% of the variance for the political participation variable. Only the socioeconomic status control variable had significant interaction with political participation ($\beta = -0.18$, $p < 0.001$) with education not significantly correlated ($\beta = -0.11$, $p = 0.09$). The results regarding the control variables show negative scores due to their inverse scaling.

Third, the covariance between work-related stress and offline social leisure was significant ($\beta = 0.12$, $p < 0.05$).

### 5.3.3. Post Hoc Analysis

Post hoc power analysis (Soper 2021) was conducted with five predicting variables and the overall explained variance of 33%. Although the sample size did not utilise the entire sample of the GSS 2014 survey, due to the ballot split, the study was sufficiently powered to detect the hypothesised relationships, as the observed power of those results is 1.

Since only one independent variable—offline social leisure—was significantly associated with political participation, mediation regression analysis was conducted to increase the internal validity of the results. Furthermore, to gain higher external validity, supplementary SEM analysis was used on the 2014 European Social Survey Sample to crosscheck the results of offline social leisure on political participation and voting. The results of those analyses results can be found in Appendices B and C.

## 6. Discussion

The first hypothesis regarding the negative relationship between work-related stress and political participation was rejected since the results were insignificant.

To answer the first part of the second research question, the paper examined the relationship between offline social leisure and political participation. The second hypothesis was accepted as offline social leisure was positively associated with social participation. For the latter part of the second research question, the paper looked at the mediating effect of participatory efficacy on the relationship between offline social leisure and political participation. The third hypothesis was accepted since a positive mediation effect of participatory efficacy was positive.

### 6.1. Work-Related Stress and Political Participation

The result has shown that work-related stress is not correlated with the political participation of US voters when controlled for education and socioeconomic status.

The potential reason for the rejection of the hypothesis is the nature of the question. The question asked by the interviewer directly requested the participant for their subjective attitude towards the frequency of their appraisal of work-related stress in their workplace (NORC n.d.b).

The social identity theory indicates that the always-improving entrepreneur, who embraces their suffering to achieve one's true potential in the meritocratic system (Elliott 2018), uses identified motivation to appraise the stressor if one identifies with the neoliberal in-group (Littler 2013). This finding was central to Meurs et al. (2010) self-categorisation model of stress, where consultancy workers appraised their stress as avenues for growth or to explore their potential, further suggesting the pervasiveness of the meritocratic injunctive norm (Littler 2013).

Therefore, the impacts of stress-inducing working demands would not be addressed by the respondents upholding the meritocratic ideal. Since 70% of Americans believe in the American dream (Younis 2021), the representative sample should reflect those attitudes similarly and refrain from primary appraisals of stress. Thus, for subjects to still be related to the in-group standard and experience autonomy and competence (Ryan and Deci 2000), they must neglect some parts of the demanding employment conditions, which are increasingly precious, as 60% of Americans are chronically stressed from work-related demands (CIGNA 2020). Otherwise, their intrinsic motivations to pursue the American dream would vanish for both entrepreneurial groups.

The results expand the research conducted by Ojeda et al. (2020) and the whole resource theory of political participation (Barrett and Brunton-Smith 2014). Although work-related stress is not correlated with lower political participation, upcoming research should allocate its resources to other types of stressors since the Ojeda et al. (2020) study has demonstrated that overall stress impacts political participation. Thus, in political psychology research, stress needs to be further examined to pinpoint the impact of particular stress promoters on psychological motivation influencing political participation.

Last, to entirely disregard the impact of work-related stress coming from the working environment, there has to be a thorough examination of more objective variables such as perceived autonomy, control, or work-related decision-making, which are objectively decreasing stress appraisals towards the work-related stressors as they induce perceived autonomy and competence (Deci et al. 2017; Lopes et al. 2013). It is necessary to identify whether the overall decline in intrinsic motivation towards collective behaviours (Wuttke 2020) is caused by the persistent challenges from the ever-changing stressful environment imposed by today's labour market, further diminishing individual cognitive resources, as the prior research on cognitive load theory suggests (Wosnitza et al. 2009).

### 6.2. Offline Social Leisure and Political Participation

The study found that offline social leisure has a direct positive correlational relationship with political participation. Although the effect size is 0.28, it still has much significance in explaining political participation since leisure time is an understudied phenomenon (Pickard 2017), and the decline in social leisure (Adorno 2016; Bartolini et al. 2011; Marcuse 1970) is evident throughout the entire labour force (Lee and Lee 2015).

Offline social leisure is more positively associated with political participation than education and socioeconomic status. Those results are even more pronounced in the regression analysis. Additionally, the SEM done on the European Social Survey affirmed this trend. Furthermore, both control variables have a significant positive relationship to offline social leisure. These results are interesting because they can partly explain why the variables are predictive in the psychological models of political participation (Cohen et al. 2001). They explain the below-average political participation of failed entrepreneurs, as

those people choose more often passive, alienating leisure activities (Lee and Lee 2015). Overall, the results confirm the importance of leisure in political psychology discourse.

The psychological effects of offline social leisure on the individual psyche explain the positive relationship. Both group-based hobbies and evenings spent with the closest circle foster social identity by promoting emotional support and a sense of belonging (Thomas et al. 2017). Those two psychological effects are the predicate variables for collective action as promoted by the dual-process model targeting the collective identity beliefs (van Zomeren et al. 2004). However, the American public is less socialising in the offline space due to varying tastes (Putnam 2020), passive leisure choices (Elliott 2018), and social media relationships offering low entry and exit costs (Williams 2007). And with the increase in loneliness rates (CIGNA 2020), the paper expects a further decline in political participation.

Those results are worrisome as offline social leisure is diminishing (Bartolini et al. 2011). With the shrinking opportunity to maintain strong ties, discussions around societal questions (Mair 2002), and the alleviation of personal resource constraints (McClurg 2003) will further decline. Additionally, the results also advance the sociological studies of Klandermans and van Stekelenburg (2013), Campbell (2013), and Schlozman et al. (2018), who described the importance of offline social groups as facilitating zones for political participation. Inside the offline social leisure groups, the motivations for political acts are being created, as political participation is overwhelmingly driven by selective social gratifications and civic gratifications (Schlozman et al. 1995). Both gratifications are embedded in the notions of collective identity, which increase the costs of nonparticipation (DiGrazia 2013). The results are especially worrying since political participatory actions are susceptible to habitual behaviour (Evans and Tilley 2017); and dependent on reinforcement, stressing the importance of measuring psychological and social motivators denoting political participation, especially when van Zomeren et al. (2008) have already mapped the relationship.

The social identity theory identifies why the individualisation process—that is in direct opposition to collective identity beliefs—continues to advance among the entire labour force. First, to be distinct in the ever-changing environment (Cushman 1990) of the labour market and maintain a stable self-concept (Peterson and Stewart 2020), the labourer cannot be closely affiliated with only one particular in-group (Vignoles et al. 2006). The always-improving entrepreneurs, who uphold the meritocratic injunctive norm (Reed et al. 2007), seek relationships valuing individual complexity and devalue the whole social network to signal occupational success (Guilbaud 2018). The failed entrepreneurs mimic those social positions since they are seen as the inferior out-group and want to escape social subordination. Therefore, they disvalue their peers and do not contribute to the community at hand (Berjot and Gillet 2011).

The loss of close relationships and the constant remarketing of oneself have a paradoxical consequence. Although the individual does not consider themselves to belong to any social group, they are still mimicking the style of multiple social groups, who still embody the capitalistic ideal (Debord 2002), but in a shallower way (Peterson and Stewart 2020) as the resource theory suggests (Barrett and Brunton-Smith 2014).

In politics, the individual chooses the party based upon a few statements since their identity is scattered through many alternatives (Hersh 2020), explaining the out-group rage justified by a few simplified points. Neither type of entrepreneur has time to be interested in politics; they do not need to engage because they lack a sense of group belonging (Stebbins 2016). The network made up of weak ties highlights system one thinking, pushing individuals to jump on the simplified versions of the out-group argument suggested by the favoured party.

The framework proposed through the social identity theory should spark new research avenues to casually confirm the positive relationship between political participation and offline social leisure. With the monetary division within the labour force and the rise of precarious working conditions (Standing 2021), the link between offline social leisure and political participation must be established.

### 6.3. Mediation Effect—Offline Social Leisure and Participatory Efficacy in Relation to Political Participation

The third hypothesis has shown that the participatory efficacies generated by group-based hobbies or evening conversations slightly increase political participation. The mediation results in the supplementary analyses mirror this effect. Although the mediating effect between offline social leisure and participatory efficacy concerning political participation was small, it was positive and significant. There is a slight correlational effect between offline social leisure and participatory efficacy. However, political participation actions are only marginally amplified by the participatory efficacies developed in social leisure settings, indicating that group-based identity and social support (Thomas et al. 2017) gathered from offline social leisure (Lim and MacGregor 2012) do not dramatically raise the belief in one's abilities to be politically active.

As all three analyses suggest, the correlation between participatory efficacy and political participation was also positive and significant, confirming the Bamberg et al. (2015) and van Zomeren et al. (2012) studies. Participatory efficacy plays a direct role in political participation at the stage of the action potential and within the act itself. These results concur with Bamberg et al. (2015), stating that participatory efficacy is the second most influential variable—after collective identity—when predicting political participation. More causal experiments have to be done to determine what variables from the SIMCA model (van Zomeren et al. 2008) are being promoted inside offline social leisure activities to precisely know why political efficacy only slightly enhances political participation via the offline social leisure pathway.

### 6.4. The Explanatory Aims

The first explanatory aim was to rule out the influence of work-related stress and offline social leisure on voting. The main analysis ruled out this relationship. Contrastingly, the supplementary SEM analysis indicates a significant relationship between offline social leisure and voting. Yet, the effect size of the correlation is minuscule. Furthermore, the covariance between voting and other forms of political participation is significant but small. The results confirm the conclusions of Bäck et al. (2011), Galais and Blais (2014), and Harder and Krosnick (2008), who empirically described voting behaviour as an act mainly caused by social norms, which is habitually reinforced. Hence, both independent variables do not aid us in explaining the voting phenomenon. The second explanatory aim was to evaluate the work-related stress/offline social leisure conceptual model. Although work-related stress is not related to political participation, offline social leisure—when controlled by the fundamental variables proposed by the resource theory of political participation—is more predictive than both control variables alone, peculiarly when the overall total effect of participatory efficacy is present. This conclusion is accurate across all three analyses.

This is also why the study did not control for other variables such as age or employment status since the purpose of the control variables was to rule out the influence of the dominant variables: education and socioeconomic status. Results surrounding the offline social leisure variable have shown that overall higher education levels and advancements in societal wealth are not the only valuable variables in studying the psychological determinants of political participation. The SEM analysis indicated that 33% of variance could be explained by the political participation variable, and the mediation regression analysis implied 37%. Those figures are acceptable since models of political participation, on average, registered 37% of the variance explained (Cohen et al. 2001).

The last explanatory aim was to examine the relationship between work-related stress and offline social leisure. This relationship was essential to establish since demands are increasing and social leisure is decreasing in American society (Hefty 2020). The results indicated a significant correlation between the two concepts. These findings are insightful as they direct future studies. Offline social leisure could be partly declining due to work-related stress. One of the possible explanations for this relationship might be the influence of work-related stress on the diminishment of social connectedness. As the Lee and Lee

study (2015) shows, those who are more stressed are choosing more atomised and passive forms of leisure activities. Although technological advancements, which enable individuals to enjoy their leisure solitarily, and the availability of products creating the diffusion of tastes and values (Bartolini et al. 2011) have especially affected the diminishment of offline social leisure, work-related stress should be further studied in relation to American leisure choices.

Overall, the statistically powered effects of offline social leisure on political participation stress the main overarching point for future studies to look at holistic societal trends beyond individual comparisons and test them to assess more environmental factors to ensure higher predictive values. Furthermore, the study results underline the need to bridge the gap between political psychology and political sociology. Political psychology has to recognise the advancements of sociological research in the realm of social networks and their crucial role in political mobilisation. Those aims highlight the thesis made by McClurg (2003), who similarly described the pitfalls of over individualised models of political participation that de-emphasise social factors and underspecify their findings. Ultimately, the study has shown the need for viewing political participation as a symbiotic relationship between individual and social approaches.

*6.5. Methodological Aim*

The final aim of this research was to propose a theoretical framework holistically explaining two significant societal trends in the US and their effects on individuals' psychological processes, partly describing the low rates of political participation in the US. The paper proposes that increasing levels of work-related stress should affect both ends of the labour force (Hefty 2020). Moreover, the conclusion made via the implementation of the self-development theory (Deci et al. 2017) and social identity theory (Iacoviello and Lorenzi-Cioldi 2019) explains the direct link towards the diminishment of collective identity regarding political participation as the SIMCA model proposes (van Zomeren et al. 2008). Either result should not discourage upcoming researchers from testing the first or second research questions since the theoretical background backs up those propositions.

*6.6. Limitations*

The first limitation of the paper lies in the latent variables. Offline social leisure was composed only of two components explaining 58% of the variance. Limited questions surrounding leisure activities, since the GSS questionnaire focuses on social attitudes and demographics (NORC n.d.a), had unfortunately narrowed the possible variables to two. Furthermore, only one variable was used to resemble the participatory efficacy, and this latent variable has a nuanced aspect to participatory efficacy, unresponsive to other overarching attitudes beyond one's group identity (Bamberg et al. 2015). Hence, another avenue for future research is to explore the predictive validity of those variables by including other items even though their face validity resembles prior studies (Bamberg et al. 2015; Lee and Lee 2015).

The second limitation examines potential causal claims, as the study used a correlational design derived from a pre-existing sample. The nature of the observational data generated every two years creates a temporal mismatch between the hypothesised paths. The discrepancy is especially notable in mediation, where the causation is difficult to infer. Hence, there should be controlled longitudinal experimental research similar to the Teney and Hanquinet study (2012), but with the usage of a representative sample ensuring an externally valid causal link between offline social leisure and social participation.

The third limitation lies in the unforeseen variable—extroversion. Extroversion might be the underlying factor behind the significant correlational results. However, the GSS representative sample accurately reflects the population split of US introverts and extroverts, which is almost even since extroverts count for 49.3% of the United States population (Myers et al. 1998). Nonetheless, the binary split between those two categories has been disputed, as 77% of Americans lie between the two distinctions (Pew Research Center 2015). The well-

cited study of Gallego and Oberski (2011) experimentally tested the influence of personality on psychological aspects related to political participation. They found nonsignificant casual results directly related to political participation activities. Nevertheless, a significant result of extroversion was recorded in the internal efficacy scores (Gallego and Oberski 2011), making political efficacy potentially susceptible to the causal interface of extroversion.

To investigate the casual relationships of offline social leisure, future research should put participants in distinct group-based hobbies and different social leisure activities with psychological variables measuring identity and socialisation to assess which parts of offline social leisure are most predictive of political participation. Next, personality traits, locus of control and collective, internal, external, and political efficiencies should be further accounted for in the proposed model of political participation.

## 7. Conclusions

Structural equation modelling has explored the hypothesised effects on the representative sample of the US voters. This paper explored the relationship of work-related stress and offline social leisure on political participation of US voters, as those two variables have transformed societal functioning over the past fifty years (Bartolini et al. 2011). The mediation effect of participatory efficacy on the relation between offline social leisure and political participation was also analysed, since participatory efficacy is the bridging variable between collective identity and collective action (Thomas et al. 2017). This study brings a unique perspective to the study of political participation since there has not been researched with representative samples focusing on offline social leisure (Lim 2008; Teney and Hanquinet 2012).

The first part of the paper found that work-related stress is not correlated with lower political participation, even though increasing demands inside the work environment lead to higher cognitive overload and diminishing intrinsic motivation (Hefty 2020). The second part of the paper examining offline social leisure found a positive association with political participation, as offline socialisation is crucial for establishing close relationships (Williams 2007) predictive of political action (van Zomeren et al. 2008). Moreover, the relationship between offline social leisure and political participation is mediated by participatory efficacy, as group-based hobbies are linked to increased beliefs in one's within-group affective beliefs (Bamberg et al. 2015).

Furthermore, the model was controlled for education and socioeconomic status, as those two variables are used as the main independent variables in the psychological models of political participation within the resource theory of political participation (Barrett and Brunton-Smith 2014). The model has indicated that offline social leisure is more predictive of political participation than the primarily used variables, especially when accompanied by the mediating relationship of participatory efficacy. The statically powered results solidified with supplementary analyses further show that psychological and sociological motivations need to be embedded in the resource theory of political participation to find a definitive answer to low political participation in the US (Pew Research Center 2018). Hence, the scientific inquiry must use a holistic approach exceeding individual differences, yet examining the broader socioeconomic phenomena affecting psychological processes.

Thereafter, the understudied aspect of offline social leisure (Pickard 2017) partly answers the inactivity paradox of political participation despite the consensus for systematic change in the United States federal government (Pew Research Center 2018). Therefore, the results and the proposed theory indicate that the political psychology discourse needs to accompany other environmental aspects to explain the factors behind the processes of those who are comparatively participating more (Campbell 2013). This paradigm shift will lead to a better understating of the underlying psychological processes denoting political participation.

**Funding:** This research received no external funding.

**Informed Consent Statement:** Patient consent was waived because it had been previously obtained by the National Opinion Research Center and the NatCen Social Research.

**Data Availability Statement:** The data presented in this study are openly available in the Center for Open Science at 10.17605/OSF.IO/9CPX2. The initial preregistration of the study is openly available in the Center for Open Science at 10.17605/OSF.IO/XM6P3.

**Conflicts of Interest:** The author declares no conflict of interest.

## Appendix A

**Table A1.** Unstandardised and Standardised Direct, Indirect, and Total Effects and Covariances of the All Paths of the Hypothesised Model.

| Path | Unstandardised Regression Weight | | | | | Standardised Regression Weight | | | | |
|---|---|---|---|---|---|---|---|---|---|---|
| | *B* | *SE* | Bootstrapping Percentile 95% CI | | | *β* | *SE* | Bootstrapping Percentile 95% CI | | |
| | | | Lower | Upper | *p* | | | Lower | Upper | *p* |
| *Direct Effects* | | | | | | | | | | |
| WRS → PP | 0.30 | 0.19 | −0.44 | 0.67 | 0.09 | 0.08 | 0.05 | 0.01 | 0.17 | 0.09 |
| WRS → VOT | −0.01 | 0.02 | −0.06 | 0.04 | 0.82 | −0.02 | 0.06 | −0.12 | 0.01 | 0.82 |
| OSL → PP | 0.69 *** | 0.13 | 0.43 | 0.94 | <0.001 | 0.28 | 0.05 | 0.17 | 0.38 | <0.001 |
| OSL → VOT | 0.03 | 0.02 | −0.01 | 0.06 | 0.09 | 0.10 | 0.06 | −0.02 | 0.22 | 0.09 |
| OSL → PE | 0.18 *** | 0.06 | 0.08 | 0.29 | <0.001 | 0.19 | 0.06 | 0.08 | 0.30 | <0.001 |
| PE → PP | 0.65 *** | 0.13 | 0.39 | 0.90 | <0.001 | 0.26 | 0.05 | 0.15 | 0.35 | <0.001 |
| PE → VOT | 0.06 *** | 0.02 | 0.02 | 0.09 | <0.01 | 0.19 | 0.06 | 0.07 | 0.31 | <0.01 |
| SEI → PP | −0.03 ** | 0.01 | −0.05 | −0.01 | <0.01 | −0.18 | 0.06 | −0.28 | −0.01 | <0.01 |
| SEI → VOT | −0.01 * | 0.01 | −0.01 | 0.00 | 0.02 | −0.16 | 0.07 | −0.29 | -0.03 | 0.02 |
| EDU → PP | −0.16 | 0.01 | −0.33 | −0.02 | 0.09 | −0.11 | 0.06 | −0.22 | 0.02 | 0.09 |
| EDU → VOT | −0.03 * | 0.01 | −0.05 | 0.00 | 0.05 | −0.14 | 0.07 | −0.29 | -0.01 | 0.05 |
| EDU → PE | −0.09 * | 0.04 | −0.15 | −0.02 | 0.02 | −0.15 | 0.06 | −0.25 | -0.03 | 0.02 |
| *Indirect Effects* | | | | | | | | | | |
| (OSL → PE) * (PE → PP) | 0.12 *** | 0.04 | 0.05 | 0.21 | <0.001 | 0.05 | 0.02 | 0.02 | 0.09 | <0.001 |
| (OSL → PE) * (PE → VOT) | 0.01 ** | 0.01 | 0.01 | 0.02 | <0.01 | 0.04 | 0.02 | 0.01 | 0.08 | <0.01 |
| *Total Effects* | | | | | | | | | | |
| (OSL → PE) * (PE → PP) + (OSL → PP) | 0.81 *** | 0.13 | 0.55 | 1.10 | <0.001 | 0.33 | 0.05 | 0.22 | 0.43 | <0.001 |
| (OSL → PE) * (PE → VOT) + (OSL → VOT) | 0.04 * | 0.02 | 0.01 | 0.07 | 0.02 | 0.14 | 0.06 | 0.02 | 0.25 | 0.02 |
| *Covariances* | | | | | | | | | | |
| WRS ↔ OSL | 0.20 * | 0.01 | 0.01 | 0.38 | 0.04 | 0.12 | 0.06 | 0.01 | 0.24 | 0.04 |
| WRS ↔ EDU | −0.22 | 0.16 | −0.49 | 0.06 | 0.11 | −0.08 | 0.05 | −0.18 | 0.22 | 0.12 |
| WRS ↔ SEI | −3.00 * | 1.39 | −5.60 | −0.67 | 0.01 | −0.13 | 0.05 | −0.24 | −0.03 | 0.02 |
| OSL ↔ EDU | −1.33 *** | 0.26 | −1.80 | −0.89 | <0.001 | 0.32 | 0.05 | −0.41 | −0.22 | <0.001 |
| OSL ↔ SEI | −8.86 *** | 2.20 | −12.88 | −4.96 | <0.001 | −0.24 | 0.05 | −0.35 | −0.14 | <0.001 |
| SEI ↔ EDU | 37.64 *** | 4.13 | 31.03 | 44.87 | <0.001 | 0.63 | 0.04 | 0.54 | 0.70 | <0.001 |

*Note.* Number of participants 295. CI = confidence interval. WRS = work-related stress. PP = political participation. OSL = offline social leisure. PE = participatory efficacy. VOT = voted in the 2012 election. SEI = socioeconomic index. EDU = highest completed school year. * $p < 0.05$. ** $p < 0.01$. *** $p < 0.001$.

## Appendix B

Mediation regression analysis for the significant relationship indicated by the SEM analysis of offline social leisure and political efficacy on political participation was applied with the bootstrapping procedure of 5000 samples via PROCESSv3.4. Prior to the analysis, all four predicting variables were assessed for independents of observations, linearity, homoscedasticity, multicollinearity, normality, and outliers regarding political participation. The relationship was controlled by socioeconomic status and education.

First, the studentised deleted residuals, leverage values, and Cook's distances were used to assess the outliers in the sample. Two participants with the ID numbers 2087 and 855 were excluded since the studentised residual indicated values greater than $-3$ standard deviations. Cook's distances were not greater than 0.09, and leverage values did not exceed 0.09. Hence, the final sample size used was 293.

The independence of observations was passed since the Durbin–Watson statistic indicated a value of 1.94. Multicollinearity was not present as tolerance values were above 0.57, and VIF showed a value that did not exceed 1.74. Linearity and homoscedasticity were examined by a scatterplot of studentised residuals over the predicted values. Furthermore, both assumptions were met. Last, the normality assumption for all three variables was met and was assessed by a Q-Q Plot.

The overall model significantly predicted political participation, $F(4, 289) = 42.15$, $p < 0.001$, $R^2 = 0.37$. Offline social leisure was statistically significant in predicting political participation ($b = 0.71$, $t[289] = 6.09$, $p < 0.001$). Offline social leisure had also a statistically significant effect on participatory efficacy ($b = 0.18$, $t[289] = 3.12$, $p < 0.01$) with the standardised coefficient of 0.19. Moreover, participatory efficacy was significant in predicting political participation ($b = 0.70$, $t[289] = 5.71$, $p < 0.001$). The indirect effect of offline social leisure on political participation through participatory efficacy was significant ($b = 0.13$, 95% CI [0.05, 0.23], $p < 0.001$) with the standardised coefficient of 0.05. Hence, the total effect of offline social leisure in political participation was statistically significant ($b = 0.86$, 95% CI [0.62, 1.11], $p < 0.001$) with the standardised coefficient of 0.37. Furthermore, the results for control variables, the regression coefficients, confidence intervals, $t$-values, standardised coefficients, and standard errors can be found in Table A2.

**Table A2.** Results of the Mediation Regression Analysis, Denoting the Relationship of All Predictors on Political Participation.

| Variable | $b$ | $SE\ B$ | $\beta$ | $p$ |
|---|---|---|---|---|
| Constant | 14.23 [11.31, 17.25] | 1.51 | – | $p < 0.001$ |
| Offline social leisure [a] | 0.74 [0.50, 0.98] | 0.12 | 0.31 | $p < 0.001$ |
| Participatory efficacy | 0.70 [0.46, 0.94] | 0.12 | 0.28 | $p < 0.001$ |
| Socioeconomic status | $-0.03$ [$-0.05$, $-0.01$] | 0.01 | $-0.19$ | $p < 0.01$ |
| Education | $-0.18$ [$-0.37$, $-0.10$] | 0.09 | $-0.12$ | $p < 0.05$ |

*Note.* The brackets denote values for the 95% confidence intervals. $R^2 = 0.37$. [a] This variable was computed from two underlying variables—social evening and social hobbies.

**Appendix C**

To assess the weight of the findings, the independent variable with significant paths—offline social leisure—was further tested on the European Social Survey (ESS) via SEM analysis. All tested pathways for the supplementary analysis are visible in Figure A1.

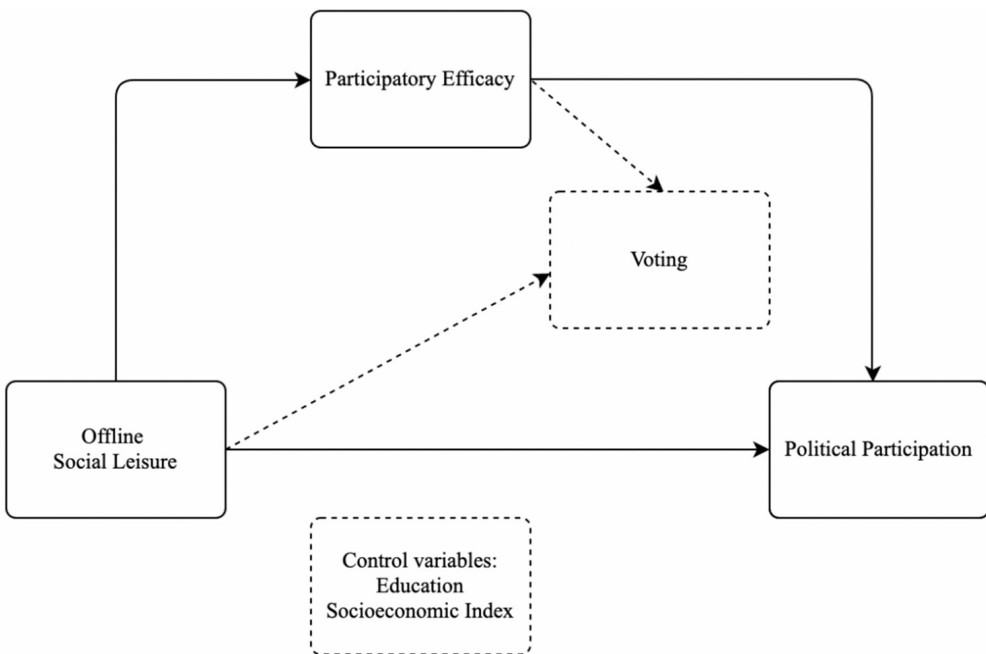

**Figure A1.** Pathway model of the relationship between offline social leisure and political participation. The political participation variable does not contain voting. The full lines denote the main analysis pathways, whereas dotted lines represent the aims for exploratory analysis.

*Appendix C.1. Methods*

Appendix C.1.1. Dataset and Sample

The strength of the ESS lies in its sample representativeness and robustness. On average, the datasets hold around 40,000 participants (Norwegian Centre for Research Data 2018). Moreover, the survey's methodology is currently viewed as the standard when conducting cross-national questionnaires (Charalampi et al. 2018). Therefore, the dataset was employed to ensure the external validity of the results as it carries a large sample size of citizens who closely resemble the American population. Since 2002, the repeated cross-national survey has used strict random probability sampling and ensures sample representativeness for most European residents older than 15 years (Norwegian Centre for Research Data 2018). The data set used represents the populations of 19 European Union member states with the addition of Norway, Israel, and Switzerland (Norwegian Centre for Research Data 2018).

The preliminary sample comprised 29,832 participants. Participants who were not eligible to vote (*n* = 1672) were excluded from the study. Last, the interquartile range method was used to identify 556 outliers within the education variable.

The final sample consisted of 27,604 participants ($M_{age}$ = 51.32, *SD* = 17.20, range: 18–114). The majority of the sample were women (52.5%). On average, the sample's highest attained year of education was 13 years ($M_{education}$ = 13.11, *SD* = 3.64, range: 4–23), and the total household income was ($M_{status}$ = 5.37, *SD* = 2.76, range: 1–10).

Appendix C.1.2. Measures

**Offline Social Leisure.** The independent variable consists of two items and represents a latent variable denoting offline social leisure. The ESS questionnaire used similar questions that represent the studies' social evenings and social hobbies variables. For the first item, the respondents were asked to indicate how often they socially meet with friends, relatives, or work colleagues (1 = *never* to 7 = *every day*). The second item was chosen to represent well the social hobbies variables: respondents were asked how often they partake in social activities compared to others of their age (1 = *much less than most* to 5 = *much more than most*).

**Participatory Efficacy.** The ESS variable mirrors the original mediation variable. The respondents were asked to indicate how confident they are in their abilities to participate in politics (0 = *not at all confident* to 10 = *completely confident*).

**Political Participation.** The first dependent variable used only those items that represented the original political participation variable. The latent variable has five items. The respondents were asked to indicate their political activity in the last 12 months regarding (1 = *yes* to 2 = *no*), "Signed a petition," "Took part in a demonstration," "Contacted politician or government official," "Displayed campaign badge," "Worked in a political party or action group". Unfortunately. the ESS dataset does not contain the same questions as the GSS dataset. Hence, the variable denoting campaign badge wearing was chosen to supplement the rally variable and the variable marking work in the political group was chosen to supplement the donation variable. Question regarding media contact had not been asked by the ESS survey. The scores were reversed coded from the original variable scores. Thus, the higher scores represent higher political participation.

**Voting**. The second dependent variable measures voting behaviour; the respondents were asked whether they had voted in the last national elections (1 = *yes* to 2 = *no*). The scores were reversed coded from the original variable.

Appendix C.1.3. Covariates

Education and socioeconomic variables were identified. The highest year of school completed was used as a proxy for education, ranging from 4 to 24 years. To assess the socioeconomic status, the variable denoting total household net income was used. The variable corresponds to the percentile distribution of all incomes gathered through the interviews (1 = *1st decile* to 10 = *10th decile*).

*Appendix C.2. Results*

Appendix C.2.1. Principal Component Analysis

For the first latent variable, political participation, the PCA looked at five underlying variables. The Kaiser–Meyer–Olkin measure was 0.70, with all individual measures greater than 0.78. Bartlett's test of sphericity was statistically significant ($p < 0.001$). Furthermore, an inspection of the correlation matrix indicated that all six variables were at least correlated above 0.33. The analysis proposed one component structure with an explained variance of 37.92%. The varimax orthogonal rotation exhibited that the adequate simple structure should retain all six variables since the average factor loading score was 0.62. All six-component loadings and communalities are visible in Table A3.

To further ensure the internal reliability of the proposed variable, Cronbach's alpha was 0.56, and the mean interitem correlations indicated a score of 0.22. Hence, the proposed latent variable could be further used in the analysis as it reliably represents 37.92% of the variance of all proposed politically participatory activities.

For the second latent variable, offline social leisure, the PCA analysis looked at two underlying variables. The Kaiser–Meyer–Olkin measure was 0.50. Bartlett's test of sphericity was statistically significant ($p < 0.001$). Furthermore, an inspection of the correlation matrix indicated they were largely correlated with a score of 0.68. The analysis proposed one component structure with an explained variance of 67.51%. The varimax orthogonal rotation exhibited that the adequate simple structure should retain all two variables since the average factor loading score was 0.82. The two-component loadings and communalities are visible in Table A3.

To further warrant the internal reliability of the proposed variable, only the mean interitem correlations could have been employed because Cronbach's Alpha cannot be used for two items (Rammstedt and Beierlein 2014). The mean interitem correlation indicated a score of 0.35. Hence, the suggested latent variable can be further used in the analysis as it reliably represents 67.51% of the variance of all proposed offline social leisure activities.

**Table A3.** Rooted Structure Matrix for PCA with Varimax Rotation of Both Latent Variables.

| Variable | Rotated Component Coefficient | |
|---|---|---|
| | Component 1 | Communalities |
| *Political Participation Latent Variable* | | |
| Worn a campaign emblem | 0.667 | 0.445 |
| Work in political group | 0.617 | 0.380 |
| Signed a petition | 0.610 | 0.372 |
| Took part in demonstration | 0.604 | 0.365 |
| Contacted politician to express view | 0.578 | 0.334 |
| *Offline Social Leisure Latent Variable* | | |
| Social hobbies | 0.822 | 0.675 |
| Social gatherings | 0.822 | 0.675 |

Appendix C.2.2. The Main Analysis

To check for multicollinearity, the variance inflation factor indicated values below 1.23, and the tolerances were above 0.81. Furthermore, Table A4 illustrates the descriptive statistics for all primary variables used in the analysis. All variables had normal univariate distributions. Therefore, all assumptions about influential points, multinormality, and multicollinearity were checked.

**Table A4.** Descriptive Statistics ($n$ = 27,604).

| Variable | *M* | *SD* | *SE* | Skewness | Kurtosis |
|---|---|---|---|---|---|
| Political participation (R) | 5.68 | 1.00 | 0.01 | 1.69 | 2.80 |
| Voted in the last election (R) | 1.78 | 0.41 | 0.01 | −1.38 | −0.09 |
| Offline social leisure | 7.51 | 2.04 | 0.01 | −0.39 | −0.22 |
| Participatory efficacy | 4.05 | 2.87 | 0.02 | 0.17 | −1.01 |
| Highest year of school completed | 13.11 | 3.64 | 0.02 | 0.06 | −0.15 |
| Total household income percentile | 5.37 | 2.76 | 0.02 | 0.07 | −1.13 |

*Note.* The reverse coded variables are denoted with an (R).

The SEM re-tested the hypothesised direct and mediating relations that were significant in the GSS sample. For the model to still be overidentified, to gain degrees of freedom, the nonsignificant household income control path for the political participation variable was deleted ($\beta = -0.01$, $p = 0.41$). Furthermore, the controlling path of education on voting ($\beta = 0.03$, $p < 0.001$) was omitted. This step is not seen problematic since the path correlation ($r(27,602) = 0.10$, $p < 0.001$) does not exceed 0.10 (Carlson and Wu 2011; Cortina et al. 2016).

Model fit indicators showed that the hypothesised model paths were good fits for the data ($\chi^2/df = 15.12/2 = 7.56$; RMSEA = 0.02; CFI = 0.99; TLI = 0.99; SRMR = 0.01). All direct paths are denoted by a standardised coefficient ($\beta$), representing the effect sizes. The main direct and exploratory paths are stated in Figure A2. All other pathways with their data encompassing unstandardised coefficients, confidence intervals, and standard errors are in Appendix D.

**Results for the Second and Third Hypothesis.** The second hypothesis had expected a direct positive relationship between offline social leisure and political participation. For this path, the analysis found a significant positive relationship ($\beta = 0.12$, $p < 0.001$).

The third hypothesis presumed a positive mediating effect between offline social leisure and participatory efficacy concerting political participation. The mediating effect is significant ($\beta = 0.05$, $p < 0.001$), with offline social leisure positively interacting with participatory efficacy ($\beta = 0.16$, $p < 0.001$), and participatory efficacy positively interacting with political participation ($\beta = 0.29$, $p < 0.001$). Moreover, the total effect of offline social leisure on political participation was also found to be significant ($\beta = 0.16$, $p < 0.001$).

**Exploratory Results**. First, offline social leisure is significantly correlated with voting ($\beta = 0.12$, $p < 0.001$). The mediating effect is significant ($\beta = 0.02$, $p < 0.001$), with offline

social leisure positively interacting with participatory efficacy ($\beta = 0.16$, $p < 0.001$), and participatory efficacy positively interacting with political participation ($\beta = 0.29$, $p < 0.001$). Moreover, the total effect of offline social leisure on political participation was also found to be significant ($\beta = 0.08$, $p < 0.001$).

Second, the proposed model fitted the data well as the model–fit indicators have shown. The model explained 15% of the variance for the political participation variable. Only the education control variables had significant interactions with political participation ($\beta = 0.12$, $p < 0.001$). Last, the correlation between political participation and voting is positive and significant ($\beta = -0.09$, $p < 0.001$).

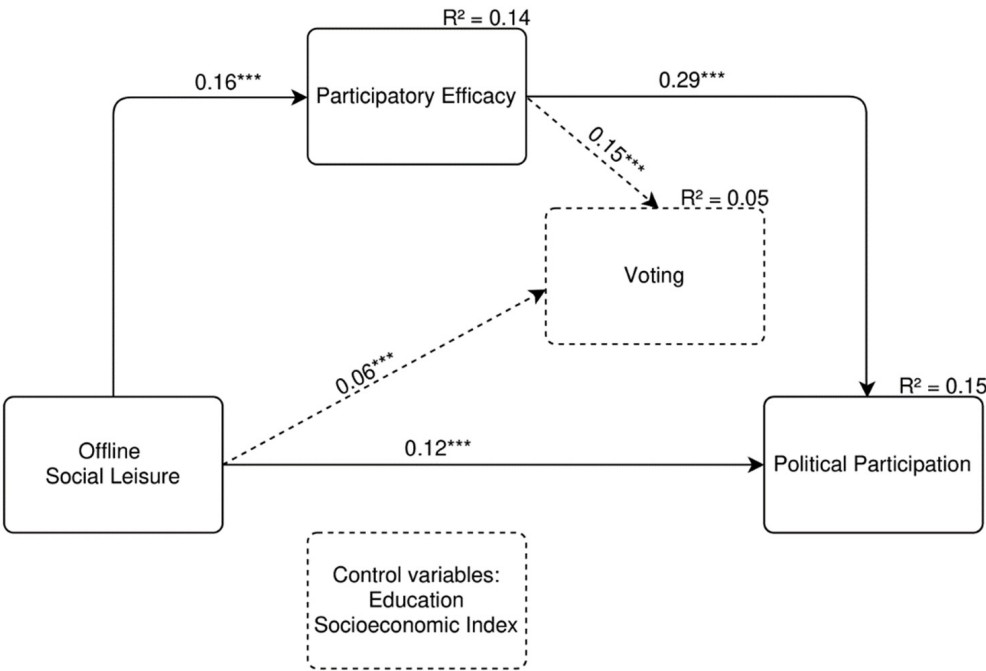

**Figure A2.** Pathways' effect sizes for the conceptual model of the relationship between offline social leisure and political participation. The political participation variable does not contain voting. The full lines denote main analysis pathways, whereas dotted lines represent exploratory analysis pathways. *** $p < 0.001$.

Appendix C.2.3. Post Hoc Analysis

Post hoc power analysis (Soper 2021), with three predicting variables and the overall explained variance of 15% for political participation, indicated observed statistical power of 1. One was also observed for the voting variable. Hence, the supplementary analysis was sufficiently powered to detect the hypothesised relationships as the observed power of those results is 1.

## Appendix D

**Table A5.** Unstandardised and Standardised Direct, Indirect, and Total Effects and Covariances of the All Paths of the Hypothesised Model Done on the European Social Survey Sample.

| Path | Unstandardised Regression Weight | | | | | | Standardised Regression Weight | | | | |
|---|---|---|---|---|---|---|---|---|---|---|---|
| | *B* | SE | Bootstrapping Percentile 95% CI | | | β | SE | Bootstrapping Percentile 95% CI | | |
| | | | Lower | Upper | *p* | | | Lower | Upper | *p* |
| Direct Effects | | | | | | | | | | |
| OSL → PP | 0.06 *** | 0.01 | 0.05 | 0.06 | <0.001 | 0.12 | 0.01 | 0.11 | 0.13 | <0.001 |
| OSL → VOT | 0.01 *** | 0.01 | 0.01 | 0.01 | <0.001 | 0.06 | 0.01 | 0.05 | 0.07 | <0.001 |
| OSL → PE | 0.22 *** | 0.01 | 0.21 | 0.24 | <0.001 | 0.16 | 0.01 | 0.15 | 0.17 | <0.001 |
| PE → PP | 0.10 *** | 0.01 | 0.10 | 0.11 | <0.001 | 0.29 | 0.01 | 0.28 | 0.30 | <0.001 |
| PE → VOT | 0.02 *** | 0.01 | 0.02 | 0.02 | <0.001 | 0.15 | 0.01 | 0.14 | 0.16 | <0.001 |
| SEI → VOT | 0.02 *** | 0.01 | 0.01 | 0.02 | <0.001 | 0.10 | 0.01 | 0.09 | 0.11 | <0.001 |
| SEI → PE | 0.13 *** | 0.01 | 0.01 | 0.02 | <0.001 | 0.13 | 0.01 | 0.11 | 0.14 | <0.001 |
| EDU → PP | 0.33 *** | 0.01 | 0.03 | 0.04 | <0.001 | 0.12 | 0.01 | 0.11 | 0.13 | <0.001 |
| EDU → PE | 0.20 *** | 0.01 | 0.19 | 0.20 | <0.001 | 0.25 | 0.01 | 0.24 | 0.26 | <0.001 |
| Indirect Effects | | | | | | | | | | |
| (OSL → PE) * (PE → PP) | 0.02 *** | 0.01 | 0.02 | 0.02 | <0.001 | 0.05 | 0.01 | 0.04 | 0.05 | <0.001 |
| (OSL → PE) * (PE → VOT) | 0.01 *** | 0.01 | 0.01 | 0.01 | <0.001 | 0.02 | 0.01 | 0.02 | 0.03 | <0.001 |
| Total Effects | | | | | | | | | | |
| (OSL → PE) * (PE → PP) + (OSL → PP) | 0.08 *** | 0.01 | 0.07 | 0.09 | <0.001 | 0.16 | 0.01 | 0.15 | 0.17 | <0.001 |
| (OSL → PE) * (PE → VOT) + (OSL → VOT) | 0.02 *** | 0.01 | 0.01 | 0.02 | <0.001 | 0.08 | 0.01 | 0.09 | 0.07 | <0.001 |
| Covariances | | | | | | | | | | |
| PP ↔ VOT | 0.03 *** | 0.01 | 0.03 | 0.04 | <0.001 | 0.09 | 0.01 | 0.10 | 0.08 | <0.001 |
| OSL ↔ EDU | 0.93 *** | 0.05 | 0.85 | 1.03 | <0.001 | 0.13 | 0.01 | 0.11 | 0.14 | <0.001 |
| OSL ↔ SEI | 0.56 *** | 0.03 | 0.49 | 0.63 | <0.001 | 0.10 | 0.01 | 0.09 | 0.11 | <0.001 |
| SEI ↔ EDU | 3.63 *** | 0.06 | 3.51 | 3.76 | <0.001 | 0.36 | 0.01 | 0.35 | 0.37 | <0.001 |

*Note.* Number of participants 27,604. CI = confidence interval. PP = political participation. OSL = offline social leisure. PE = participatory efficacy. VOT = voted in the last election. SEI = household income. EDU = highest completed school year. *** *p* < 0.001.

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
