# Peer review of "Relationship of Work-Related Stress and Offline Social Leisure on Political Participation of Voters in the United States"

_socsci, doi:10.3390/socsci11050206_

Round 1

Reviewer 1 Report

This work addresses a very interesting topic. The authors rightly observe that current psychological models that explain political participation are studied through education and socioeconomic status; And as these assumptions say, they cannot explain the low general rates of political participation.

Their approach is well thought out.

The theoretical framework is complete and robust, synthetically addressing different theories on which to base its analysis and discussion.

The research methodology is perfectly planned, both the independent and dependent control variables, hypotheses, etc.

On the other hand, the data and statistics clearly show the results and the validity of the work. The discussion is brilliant.

I congratulate the authors.

-First: The selection of the topic. The authors rightly observe that current psychological models that explain political participation are studied through education and socioeconomic status. For a long time nobody has openned new  perspectives to study the topic; and as the author explains they education and socioeconomic status don't  explain the low general rates of political participation.

- Second: the literature review.  The theoretical framework is complete and robust, synthetically addressing different theories on which to base its analysis and discussion.

- Third: Methodology: Their approach is well thought out. The research methodology is perfectly planned, both the independent and dependent control variables, hypotheses, etc.

- Fourth: Results. The data and statistics clearly show the results and the validity of the work. 

- And finally: The discussion is brilliant.

Author Response

Thank you for your encouraging words. Is there anything worth editing?

Reviewer 2 Report

This manuscript falls short on all dimensions.

The theory is quite convoluted. I never could sink my teeth into the argument. The author tries to weave too many competing claims and theories into their argument. I never got a sense of what the AUTHOR was trying to argue. Instead, it was a messy web of what others have said. Operationalization is unclear because the author weaves methods and theory together. The terms used are confusing.

The language is way too unclear. For example, “In neoliberal society, work is regarded as an emancipatory process of cultivating one’s happiness by weighing one’s contributions to the advancement of the neoliberal society”…or…”All in all, work in American society has become the primary source of merit and the means for pursuing one’s liberation from which one appraises their position to the collective.” These sentences are unnecessary and confusing.

The methods are also confusing. Way too many observations are eliminated. An N of 299 is just not sufficient. Correlation does not tell us much either.

Author Response

Dear Sir or Madam, 

The reason why the sample size is 299 is that the GSS survey does not have questions about social leisure and work-related stress in their core questionnaire. For example, the social leisure questions were only asked in the 2014 issue make it the more recent data unavailable. The GSS divided those unique questions into three categories and distributed them among participants. This is why only 299 were at the end available because work-related stress and social leisure questions were in a different category. Furthermore, this is also why SEM has been used analysis as the observed power of the results is 1. However, indeed some imputation methods could have been used. The correlational design was a standard in the political psychology studies that looked at political participation. Could you give me some guidance on how would you like to see the methods sections and what are the problems with references?

Thank you for the time given to this paper

Reviewer 3 Report

This is an interesting observational study that evaluates the relationships between political participation and (1) work related stress and (2) social leisure activity. I think the paper is potentially publishable, but I would like to see the following:

  1. Though the literature reviewed is extensive, hardly any of it is in the area of American political participation. There are a lot of relevant studies of the factors that explain participation that are not related to education/SES. Lots and lots of earlier work on the roles of social identity/group consciousness, social pressure, and the. like. I recommend the authors consult a relevant Oxford Handbook chapter or Annual Review of Political science entry to beef this up.
  2. They need to define neoliberalism. I know what they are talking about but they cannot just assume that all readers do.
  3. They talk a lot about political participation without mentioning voting. I would clarify from the outset that the study is about non-voting forms of participation.
  4. They need to acknowledge that there is not really much of a paradox between the growth in education over time and participation rates. The latter have been growing. The past two electoral cycles witnessed the highest turnout rates in US history.
  5. They have no evidence to support the first hypothesis so unless they really want to take a stand on that null finding, I would drop it and focus on the second hypothesis, which will have the benefit of shortening the theory section--which is needed.
  6. In so doing, they can also evaluate H2 as it relates to voting. That is frankly the most important form of political participation and they should evaluate it.
  7. I would like to see the results as they pertain to each item in the outcome variable index.
  8. I would like to see regular regression results. SEM is extremely sensitive to specific modeling decisions, especially when one estimates the measurement model and the regression model simultaneously
  9. I would get rid of the efficacy intermediary. In some ways, this is just another item that could be considered a DV, because it explicitly asks how important they consider POLITICAL activity to be.
  10. In some ways, both sides of the equation are measuring much the same thing, given that doing things like attending rallies and demonstrations are in fact social leisure activities. They need to acknowledge that as a shortcoming, and acknowledge that the relationship they capture could be a byproduct of something like extraversion
  11. Most importantly, a sample size of 299 when they started out over 2k is unacceptable. I assume that the GSS asked these questions in lots of different years. Why focus on 2014? They should merge a bunch of years together (and include year dummies) to not only boost the N but also demonstrate robustness over time.

Author Response

Dear Sir or Madam, 

Thank you for your thorough review. we agree with points 1-4 and 7, we integrate them into the new version. For the fifth point, we believe that publishing a null hypothesis is also a valuable outcome for future research. However, if the theoretical part in your opinion has to be shorter then this point is a valid one. Sixth, do you then propose to add voting as a new variable? Eight, the regression analysis was previously in the manuscript but it was taken out as the results were comparable to shorten the manuscript. Ninth, this point is definitely up to debate. Tenth, even though there should be only a slight overlap between those two variables only due to the social part of the social groups it will be addressed in the limitations - thank you for noticing it.

Last, The reason why the sample size is 299 is that the GSS survey does not have questions about social leisure and work-related stress in their core questionnaire. For example, the social leisure questions were only asked in the 2014 issue (as you can see here: https://gssdataexplorer.norc.org/variables/4931/vshow) making the more recent data unavailable. The GSS divided those unique questions into three categories and distributed them among participants. This is why only 299 were at the end available because work-related stress and social leisure questions were in a different category. However, after some months of maintenance, the European Social Survey is again available. I have run a quick linear regression analysis with Social leisure and political efficacy as DV controlling for education and household income. And they both outperform the controlling variables. It was run on 35k participants. This could give the results more external validity if it is combined with GSS. The only downside is that 2 variables from political participation are missing, the donation and political rally participation. But those things can also be supplemented. What do you think? 

Those are the ESS questions:
Able to take active role in political group
Contacted politician or government official last 12 months
Signed petition last 12 months
Taken part in lawful public demonstration last 12 months
Posted or shared anything about politics online last 12 months
How often socially meet with friends, relatives or colleagues
Take part in social activities compared to others of same age
Highest level of education
Household's total net income, all sources

We are very grateful for your guidance and the time is given to this paper

Round 2

Reviewer 3 Report

This manuscript is much improved, in light of the revisions the author(s) have made. I still think they need to add turnout as an outcome variable, and yes, they should supplement with the additional data they mention (at least in supplementary materials). Ditto re: the regression analysis (in supplementary materials online, not in the main text).

Author Response

The manuscript was updated mainly to include voting as another outcome variable and to include another SEM analysis of the 2014 European Social Survey dataset (N = 27604) to boost the study's external validity. All those changes have greatly improved the manuscript; especially, the incision of voting and the ESS sample made the manuscript more sound and relevant. The team want to thank you for your thorough work.